# Shearo-caloric effect enhances elastocaloric responses in polymer composites for solid-state cooling

Shixian Zhang [1,2,6], Yuheng Fu[1,6], Xinxing Nie[1], Chenjian Li[1], Youshuang Zhou[3], Yaqi Wang[1], Juan Yi[1,4], Wenlai Xia[1], Yiheng Song[1], Qi Li [4], Chuanxi Xiong [1,7] ✉, Suxin Qian [5] ✉, Quanling Yang [1] ✉ & Qing Wang [2,7] ✉

Room-temperature elastocaloric cooling is considered as a zero-global-warming-potential alternative to conventional vapor-compression refrigeration technology. However, the limited entropy and large-deformation features of elastocaloric polymers hinder the creation of the breakthrough in their caloric responses and device development. Herein, we report that the addition of a small amount of inorganic nanofillers into the polymer induces the aggregate of the effective elastic chains via shearing the interlaminar molecular chains, which provides an additional contribution to the entropy in elastocaloric polymers. Consequently, the adiabatic temperature change of −18.0 K and the isothermal entropy change of 187.4 J kg$^{-1}$ K$^{-1}$ achieved in the polymer nanocomposites outperform those of current elastocaloric polymers. Moreover, a large-deformation cooling system with a work recovery efficiency of 56.3% is demonstrated. This work opens a new avenue for the development of high-performance elastocaloric polymers and prototypes for solid-state cooling applications.

Caloric effects (CEs) refer to the nominally reversible thermal responses of materials under the application and removal of external mechanical (mechanocaloric effect), magnetic (magnetocaloric effect)[1,2] and electric (electrocaloric effect)[3,4] fields, and are typically parameterized as the adiabatic temperature change ($\Delta T_{adi}$) and the isothermal entropy change ($\Delta S_{iso}$)[5–7]. Extensive research has been conducted on the caloric effects for solid-state cooling over the past two decades[8,9]. At present, the most developed mechanocaloric materials are shape-memory-alloys with the martensitic transitions and comprised of elastocaloric materials driven by uniaxial force, and barocaloric materials operating under hydrostatic pressure[6]. The thermal responses to the multiple external fields, due to the interplay

in material structures, and electric and magnetic polarizations, spur further interest[10,11]. Among them, elastocaloric effects (e-CEs) cooling technologies have been considered as the most promising alternative for near room-temperature cooling technologies owing to the rapid advance in the development of thermoelastic materials and their prototypes[12–15].

In recent years, the e-CE polymers with phase transitions, e.g., the strain-induced crystallization in natural rubber (NR) and the α-β ferroelectric transition in poly(vinylidene fluoride-trifluoroethylene-chlorotrifluoroethylene) terpolymer, along with their cooling devices, have received renewed attention due to their unique features of rubber elasticity, low driving stress, high fatigue life, recyclability and

[1]State Key Laboratory of Silicate Materials for Architectures, and School of Materials Science and Engineering, Wuhan University of Technology, Wuhan, China. [2]Department of Materials Science and Engineering, The Pennsylvania State University, University Park, PA, USA. [3]School of Materials Science and Engineering, Hubei University, Wuhan, China. [4]Department of Electrical Engineering, Tsinghua University, Beijing, China. [5]Department of Refrigeration and Cryogenic Engineering, School of Energy and Power Engineering, Xi'an Jiaotong University, Xi'an, Shaanxi, China. [6]These authors contributed equally: Shixian Zhang, Yuheng Fu. [7]These authors jointly supervised this work: Chuanxi Xiong, Qing Wang. ✉e-mail: cxiong@whut.edu.cn; qiansuxin@xjtu.edu.cn; yangql@whut.edu.cn; wang@matse.psu.edu

processability[16–25]. e-CEs have also been developed in the polymer composites to take advantage of the reinforcement, nucleation ability, and high thermal conductivity of fillers[26–28]. More universally, the ubiquitous conformational entropy changes, when the polymer chains transition from curled to straight during polymer deformation, endow polymers with large e-CEs[29–31]. However, the entropy change in pristine polymers, including lattice energy and conformational changes, is not sufficient under uniaxial force, which limits the performance of current e-CE polymers. For example, molecular chains in a polymer network parallel to the stretching direction tend to be oriented under uniaxial stress and act as an entropy reservoir to produce e-CE, while those perpendicular to the stretching direction are unable to undergo large conformational changes and thus have minimal contribution to the entropy change. On the other hand, solid-state cooling critically depends on functional development of the cooling devices that are tailored to solid refrigerant materials and their responses to the external fields[14,15,32–37]. However, the inherent large-deformation characteristics of polymers limit the development of their e-CE cooling devices due to the demand for large driving strokes. Only a few works have recently been reported on the large-deformation cooling polymer systems[21,22,29,38]. For example, by harnessing snap-through instability in the soft capacitor to drive the expansion and contraction of an NR membrane balloon, a high operating frequency function was achieved for the e-CE cooling device[21]. Another type of large-deformation cooling device utilized the geometric design of NR tubes to endow the device with a thermal conductivity compensation function[22,38]. Nevertheless, the energy recovery has not yet been realized in the large-deformation systems, which is crucial for further improving the system efficiency[39].

The mechanical field can be divided into uniaxial tension or compression (e-CE), shear, bending, and torsion[40,41]. The practical uniaxial force when applied to the fundamental elastic element inside polymer materials can be the combination of these four basic forms of stresses and deformations. We thus hypothesize that the purposeful combination and optimization of these basic forms of deformations are effective to increase the entropy change of the CEs.

In this work, carbon nanofillers are used as the shear layers to reconfigure the elastic network of polystyrene-b-poly(ethylene-co-butylene)-b-polystyrene (SEBS) and induce the formation of elastic elements with the coexistence of multi-entropy reservoirs. The thermal response generated by the additional entropy, which refers to the molecular chains bonded between the interlayer of the nanofillers and is driven by the relative displacement of the nanofillers, is described as the shearo-caloric effect (s-CE). Consequently, the total e-CE $\Delta T_{adi}$ benefits from both additional s-CE and original e-CE of the SEBS chains and has been directly measured with an infrared thermometer in a uniaxial-strain-controlled system under indoor environment (Details in Supplementary Methods and Supplementary Fig. 1). Furthermore, a double-unit cooling device with the work recovery capability is designed and demonstrated for the polymer refrigerant with the large-deformation characteristics.

## Results

### CE in SEBS composites

The fabrication and labeling of the polymer composites are summarized in Supplementary Methods, Supplementary Table 1, and Supplementary Figs. 2–4. The SEBSs with high molecular chain-length uniformity[29] and high orientation ability[30] are labeled as HHs. The molecular weight distribution range, the side group content, and the weight fraction of polystyrene (PS) of HHs are 79,000 g mol$^{-1}$, 11 mol%, and 29 wt.%, respectively. The HHs composites filled with the graphene nanoplatelets (GNS) are labeled as GNS/HHs. As shown in Fig. 1a, the highest $|\Delta T_{adi}|$ value of GNS/HHs in the cooling process, which was measured under the largest strain ($\varepsilon$) of 600%, increases initially and then decreases with the increase of the GNS loading. The $|\Delta T_{adi}|$ reaches the maximum value of 18.0 K at 1 wt.% GNS (1GNS/HH). The infrared measurements (Fig. 1b, c) verify the temperature increase of 1GNS/HH during the stretching process (P1-P2), the heat transfer

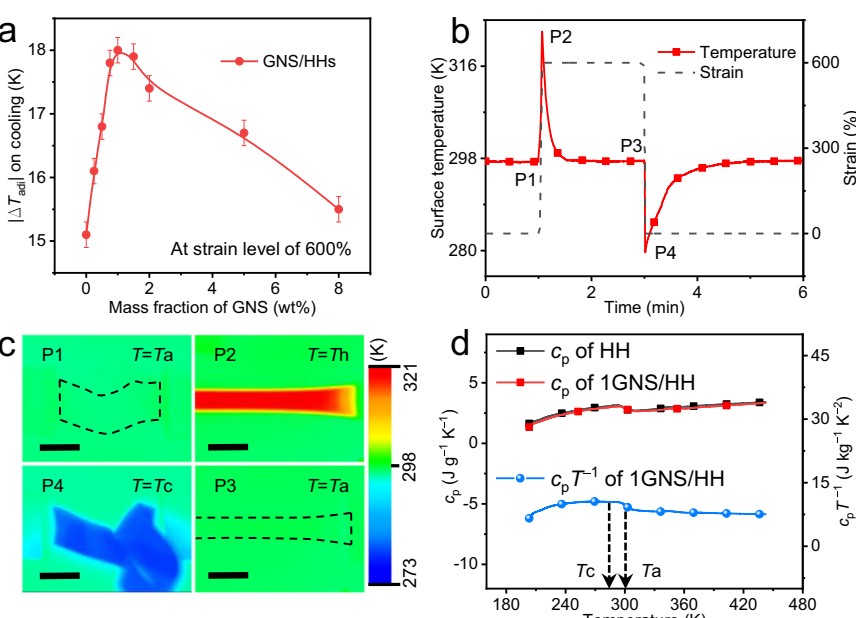

**Fig. 1 | e-CE cycle and temperature change of GNS/HHs. a** $|\Delta T_{adi}|$ of GNS/HHs on cooling process. All data were collected at a strain level of 600%. The error bars represent the average values of 6 tests under the same conditions. **b** Typical strain evolution (analogized from the displacement of the slide table) and the resulting temperature variation of 1GNS/HH during a single e-CE cycle. During P1–P2, the sample is stretched adiabatically by a uniaxial tension, increasing the surface temperature ($T_a$ to $T_h$). During P2–P3, the strain remains constant, and the thermal entropy decreases isothermally until $T = T_a$. During P3–P4, the specimen retracts adiabatically with the unloading of the tension, further decreasing the temperature of the specimen ($T_a$ to $T_c$). During P4–P1, the thermal entropy of the material increases until $T = T_a$. **c** Infrared thermal images of 1GNS/HH at different processes were recorded using an infrared thermal imager. The dashed line describes the outlines of the sample. The scale bar is 20 mm. **d** The specific heat versus temperature curves tested by DSC.

process with the environment (P2-P3 and P4-P1), and the decrease of $|\Delta T_{adi}|$ upon recovery of strain (P3-P4). The $|\Delta T_{adi}|$ values of the selected GNS/HHs at different strain levels during the cooling process are summarized in Supplementary Fig. 5. Owing to the small content of GNSs in the polymer composites, the specific heat ($c_p$) of 1GNS/HH measured by differential scanning calorimetry (DSC) is similar to that of HH matrix (Fig. 1d). Therefore, the different $\Delta S_{iso}$ values between these two samples during the heat equilibration process on cooling (P4-P1) are mainly related to the $|\Delta T_{adi}|$ during P3 to P4 (details of the energy conversion relationship of e-CE refer to Supplementary Discussion 1). The $\Delta S_{iso}$ of 1GNS/HH on cooling is estimated at 187.4 J kg$^{-1}$ K$^{-1}$ by integrating $c_p$ $T^{-1}$ from $T_c$ to $T_a$ (Fig. 1d) according to Supplementary Equation (7), where $T$ is the real-time temperature of the specimen, $T_c$ is the sample cooling-temperature at P4 and $T_a$ corresponds to the initial temperature of the sample at P1. Notably, the room temperature e-CE obtained in this work exceeds those of current elastocaloric polymers as shown in Supplementary Fig. 6. The coefficient of performance (COP$_{mat}$) of 1GNS/HH is about 15.7 during stable e-CE cycles (details in Supplementary Discussion 2).

## Structure evolution and enhancement mechanism of CE for GNS/HHs

The e-CE of 1GNS/HH is about 20% higher than that of HH, which can be attributed to the s-CE as a result of the reinforcement of the fundamental elastic elements in the polymer composites. As illustrated in Fig. 2a and Supplementary Fig. 9, the GNS/HHs composites contain two kinds of fundamental elastic units, which present in the matrix without the GNS nanofillers (State I) and are integrated with the GNS nanofillers (State II), respectively. At the initial state, as shown in the transmission electron microscopic (TEM) images (Fig. 2b, c) of 1GNS/HH, there appear two transition layers with a thickness of about 2 nm on both sides of the surfaces of GNS, indicating intimate arrangements of the polymer chains in the transition layers as a consequence of the adsorption affinity introduced by the nanofillers. In addition, the π−π interactions formed between the PS domains in the polymer and the GNS nanofillers enhance the interfacial adhesion between the nanofillers and the polymer matrix[42,43]. The polymer matrix thereby wets the surface of GNS, resulting in a more compact structure on the surface of nanofillers. The influence of GNSs on the microdomain structures of the polymer composites has been investigated by atomic force microscopy (AFM) (Fig. 2d–g). The microphase-separated morphology with the cylindrical PS domain of a diameter of about 20 nm dispersed in the poly(ethylene-co-butylene) (PEB) continuous phase are observed in both the composite (Fig. 2e) and the neat matrix (Fig. 2g). Compared to a continuous morphology of the PS blocks in HH (Fig. 2g), the PS blocks in 1GNS/HH display a more dissociated morphology and a larger number of crosslinking points per unit volume because the PS segments at the polymer chain end interact with GNS via π−π interactions and thus form smaller PS domains on the surface of GNS (Fig. 2e). The PS segments at the other end of the polymer chains would extend into the matrix and interact with another GNS and form the physical crosslinking domains in the matrix, thus realizing the stress transfer of the entire composite network. Lower magnifications of the TEM image and AFM phase diagram of 1GNS/HH are displayed in Supplementary Fig. 10. At the initial state, the two-dimensional wide-angle X-ray diffraction (2D WAXD) patterns of HH (Fig. 2h) and 1GNS/HH (Fig. 2i) show wide dispersion rings near $q = 1.3$ Å$^{-1}$, indicative of random coil structures of the molecular chains in the matrix. The clear diffraction ring (Fig. 2i) corresponding to the random-oriented (002) crystal plane of GNS appears at $q = 1.87$ Å$^{-1}$, indicating the random distribution of GNS nanofillers in the composite. With the increase of the GNS content, the relative intensity corresponding to (002) and (100) crystal planes gradually increases, while the width of the amorphous peaks at $q = 1.3$ Å$^{-1}$ remains constant

(Fig. 2k). This suggests that, although the film morphologies of HH and 1GNS/HH are somewhat different, the randomness of chain arrangement and the degree of short-range order of the polymer chains are unaffected by the GNSs loading, therefore leading to the same conformational entropy of HH and 1GNS/HH at the initial state.

At the elongation state of the composites, the relative intensities of the dispersion ring corresponding to the random molecular chains are concentrated perpendicular to the stretching direction (Fig. 2h–j). This indicates that, during the transition from P1 to P2, the PEB elastic blocks undergo a transition from random curled chain conformations to oriented chain conformations along the elongation direction, resulting in the reduction of conformational entropy ($\Delta S_\lambda < 0$). The intensity of the diffraction ring corresponding to the (002) crystal plane of GNS is also enhanced perpendicular to the stretching direction (Fig. 2j), suggesting that the GNSs is aligned parallel to the stretching direction due to the traction of the interacting polymer chains. As illustrated in Fig. 2l, the full-width half maximum (FWHM) of the azimuthal ($\Psi$) integral curves for GNS/HHs after deformation gradually decreases with increasing the GNSs loading from 0 to 1 wt%[29,30,44]. According to the Hermans' equation, the polymer chains in 1GNS/HHs display a higher degree of orientation than those in the pristine polymer and the composites filled with lower content of GNSs during deformation, thus resulting in a higher $\Delta S_\lambda$ value. This is mainly due to the shear displacements of GNSs in the polymer composites induced by the shear elongation of the π−π interaction-bonded interlaminar molecular chains during deformation (state II in Fig. 2a). Comparatively, the polymer chains without the shear effect of GNSs remain curly during deformation (state I in Fig. 2a). When the polymer composite returns to the initial length, the nanofillers and the increased elastic molecular chains recover to the original random distribution state as confirmed in the 2D WAXD patterns (Supplementary Fig. 13).

The above processes were further validated through the dynamic simulations (details in Supplementary Methods). As illustrated in Supplementary Fig. 9, the molecular chains in the selected fundamental elastic units for both HH and 1GNS/HH exhibit random curled chain conformations at the initial state. When elongated, the shear dislocation of GNSs caused by the stress transfer in the composite network results in an extension of the bonded interlayer molecular chains (Supplementary Fig. 9h). Hence, the interlayer chains in 1GNS/HH are more outstretched along the deformation direction at the elongation state, which is equivalent to increasing the density of effective elastic chains. Accordingly, the total end-to-end distance of the chains in the fundamental elastic unit in 1GNS/HH is greater than that in HH. The statistic images (Fig. 2m) for the end-to-end distance of the above HH and GNS/HH models show that the molecular chains of GNS/HH possess a similar mean square end-to-end distance value of about 4054 Å$^2$ to HH at the initial state. While at the elongation state, the end-to-end distance of 1GNS/HH is about 91,474 Å$^2$, higher than 75,707 Å$^2$ of HH, which indicates that there are more stretched polymer chains in 1GNS/HH. The increase in the number of effective elastic chains of the polymer composite denotes that there are more molecular chains participating in the conformational change process during deformation, thus resulting in enhanced s-CE, and consequently, the large total e-CE responses. It can be considered that the contribution of the e-CE in HH mainly comes from the effective elastic chains of the polymer matrix, while the total e-CE in the 1GNS/HH composite has the contributions from both the e-CE of the matrix and the additional s-CE introduced by the nanofillers. Further analysis of the quantitative contribution of these two mechanisms from the statistical results shows that the s-CE increases the e-CE of the polymer matrix by about 22% (details in Supplementary Discussion 1).

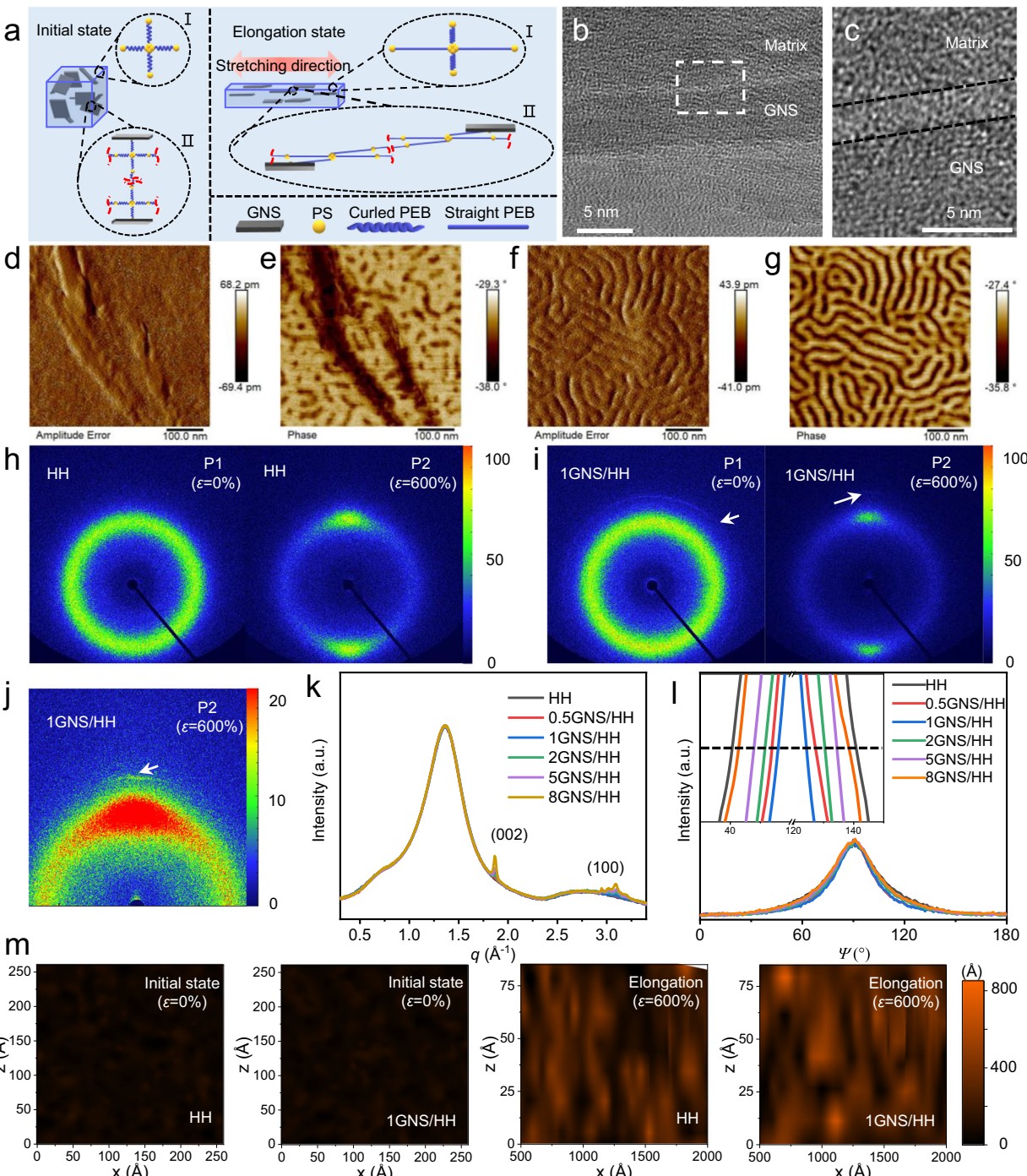

**Fig. 2 | The structure evolution and the enhancement mechanism of s-CE for GNS/HHs. a** Schematic diagram of composites undergoing deformation. States I and II represent the fundamental elastic element without nanofillers and with nanofillers, respectively. The red dashed line omits the connected repetitive elements. **b** TEM images of the ultrathin section specimen of 1GNS/HH. The darker area in the middle corresponds to the section of GNS. **c** Enlarged view of the boundary (middle region) between nanofiller (bottom region) and polymer matrix (top region) in (**b**). These three regions are divided by the black dotted lines. Amplitude image (**d**), phase image (**e**) of a selected area of 1GNS/HH and amplitude image (**f**), phase image (**g**) of a selected area of HH, measured by AFM. 2D WAXD images of HH (**h**) and 1GNS/ HH (**i**) under P1 and P2. **j** Magnification image for 1GNS/HH under P2 with enhanced contrast. The white arrows point to the (002) crystal plane of GNS. **k** 1D WAXD profiles of GNS/HHs obtained by full-tilted circular integration of the corresponding 2D WAXD patterns at P1 ($\varepsilon = 0\%$). **l** Azimuthal integral of the corresponding 2D WAXD patterns of GNS/HHs at P2 ($\varepsilon = 600\%$) obtained near $q = 1.3\ \text{Å}^{-1}$. The inset profile is a partially enlarged view of (**l**). The black dotted line indicates the position of half maximum. **m** The distribution of the end-to-end distance of GNS/HHs before and after elongation. The x and z axes represent the centroid coordinates of the molecular chains, and the color scale represents the end-to-end distance.

## Shear-damping transition in different composites

As shown in Fig. 1a, $|\Delta T_{\text{adi}}|$ exhibits a progressive rise with the GNS loading and reaches its peak value around 1 wt.% loading (i.e., the critical loading) of GNS. With further increasing the GNS content beyond the critical loading, the decrease of $|\Delta T_{\text{adi}}|$ primarily arises from the occurrence of a shear-damping transition within the reconstructed network.

As illustrated in Fig. 3a, the tan$\delta$ peak near 235 K is attributed to the glass transition temperature ($T_{\text{g}}$) of the PEB elastic blocks ($T_{\text{g}}$-PEB),

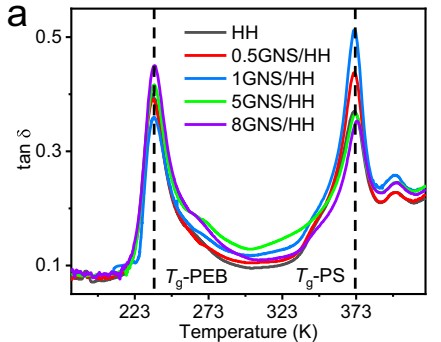 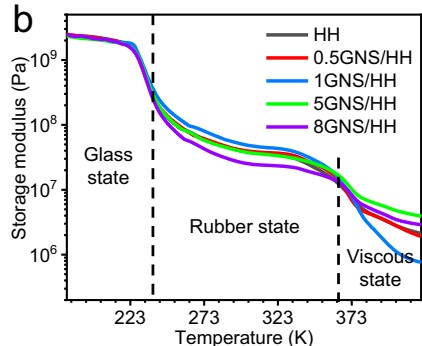

**Fig. 3 | Relationship between nanofillers loading and *s*-CE. a** Tan $\delta$ versus temperature curves of GNS/HHs. The black dotted lines indicate the peak positions corresponding to $T_g$-PEB and $T_g$-PS. **b** Storage modulus versus temperature curves of GNS/HH.

and the peak near 371 K is assigned to the $T_g$ of PS hard blocks ($T_g$-PS). Before reaching the critical loading, the peak value of tan$\delta$ corresponding to the $T_g$-PEB gradually decreases with the GNS content, while the peak assigned to the $T_g$-PS steadily increases. This indicates that the fraction of the polymer segments affected by the interfacial bonding of GNS is significantly increased, which endows the composite with more *s*-CE elastic elements, specifically, higher storage modulus under rubber state (Fig. 3b). It can be considered that the viscoelastic behavior of the reconstructed network, which is caused by the introduced GNS, is mainly reflected by the increased fraction of *s*-CE elastic elements before the critical loading. The increase of *s*-CE elastic elements reduces the content of viscous fraction and transmits stress throughout the entire cross-linking composite network, thereby attenuating the damping performance of PEB soft domains and manifesting a reduced tan$\delta$ associated with the $T_g$-PEB. On the other hand, the physical entanglements of the PS phase are weakened due to the additional π−π interactions between the PS segments and GNS, resulting in higher segmental mobility of PS during its glass transition process and leading to an enhanced tan$\delta$ associated with the $T_g$-PS[45]. Therefore, the degree of orientation (Fig. 2l) and the resulting *e*-CE (Fig. 1a) increase with the GNS loading up to 1 wt.%.

With the further increase of GNS loading (>1 wt.%), both the degree of orientation of the deformed composite and the obtained *e*-CE decrease. This is owing to the fact that the partially reconstructed network could undergo a shear-damping transition and exhibit higher damping performance (Fig. 3a) when the GNS content is beyond the critical loading. The increased loading of GNS reduces the interparticle distance and forms nanofiller aggregates as verified in the field emission-scanning electron microscopic (FE-SEM) images (Supplementary Fig. 14). The formation of the nanofiller aggregate network greatly reduces the relative movement between GNSs and the associated shear effects, which means that the deformation or destruction of GNS aggregates under dynamic force would lead to more friction and viscosity of the embedded PEB segments[46]. The nanofiller aggregates can be considered as damping elements and contribute to the vibration absorption behavior of the reconstructed network. Thus, the deformation or destruction of the nanofiller aggregate network would significantly increase the tan$\delta$ near the $T_g$-PEB and result in a lower storage modulus in the rubber state (Fig. 3b). Moreover, a large amount of GNSs may refine the domain size of the PS phase. Consequently, a higher fraction of the PS segments could interact with GNS, thereby reducing the mobility of PS segments, resulting in smaller tan$\delta$ near the $T_g$-PS.

Interestingly, the *s*-CE and the shear-damping transition have been found in other nanofiller-polymer composites (details in Supplementary Discussion 3 and Supplementary Fig. 15), demonstrating the generality of *s*-CE. It is demonstrated that the *s*-CE could enhance the original *e*-CE of the polymer matrix, with weak chain orientation

mobility, by 30% at most. Consistent to the results of the 1GNS/HH composite, the shear-damping transition of the composites filled with a high nanofiller aspect ratio occurs at a low nanofiller loading.

**Double-unit cooling device with the work recovery capability**
In view of the *s*-CE and *e*-CE that can be generated synchronously under the same uniaxial tensile system, a uniaxial tensile driven double-unit cooling device with the work recovery capability is developed by using 1GNS/HH as a refrigerant (S1). As shown in Fig. 4a, the end of the refrigerant is fixed at the bottom of the cooling unit (U), while the other end is connected to ultra-high molecular weight polyethylene fiber (S2) through heat sealing. S2 fiber is wrapped around the driving wheel (D), and the rotation process of the driving wheel can perform the winding and unwinding process of S2 fiber, thereby achieving the stretching and recovery processes of the refrigerant. The two driving wheels (D1 and D2) of the double-unit cooling device are coaxial, which allow the torque transmission between these two cooling units (U1 and U2). Details of the preparation of the refrigerant and device are described in Supplementary Discussion 4 and Supplementary Fig. 16. At the initial state (P1), the refrigerant maintains its original length in cooling unit U1 and is elongated with a strain of 600% in cooling unit U2. During the transition from P1 to P2, the refrigerant in U1 is adiabatically stretched to a strain level of 600%, and adiabatically recovers to its initial length in U2, consequently generating heating in U1 and cooling in U2. During the transition from P2 to P3, water as heat transfer fluid is pumped between the baffle plates and refrigerants driven by the one-way pumps (V1 to V4), thus simultaneously exporting the thermal energy generated in U1 for the heat exchange in the heat sink and the thermal energy generated in U2 for cooling of the heat source. After thermal equilibrium (P3 to P4), the reverse rotation of the motor generates the adiabatic recovery of the refrigerant in U1 and the adiabatic stretching of the refrigerant in U2. During the transition from P4 to P1, one-way pumps (V5 to V8) export the cooling energy generated in U1 to the heat source and the heat generated in U2 to the heat sink.

Additionally, a single-unit cooling device is fabricated by disconnecting the coupling between cooling unit U1 and the right-angle conversion gearbox, which is used for comparing the overall performance with that of the double-unit cooling device at a strain range of 100% to 500%. When the heat sink is stable around room temperature and the heat source is maintained in an insulated reservoir, the zero power $T_{span}$ of the single-unit cooling device reaches 3.5 K after about 4800 s of cooling cycles, whereas $T_{span}$ of the double-unit cooling device achieves 3.7 K after about 2500 s of cooling cycles (Fig. 4b). The duration required for the double-unit device to achieve a $T_{span}$ balance is significantly reduced, which is mainly due to the fact that the cold water is delivered to the insulated reservoir during both transitions of P2 to P3 and P4 to P1. In contrast, the single-unit device can only be

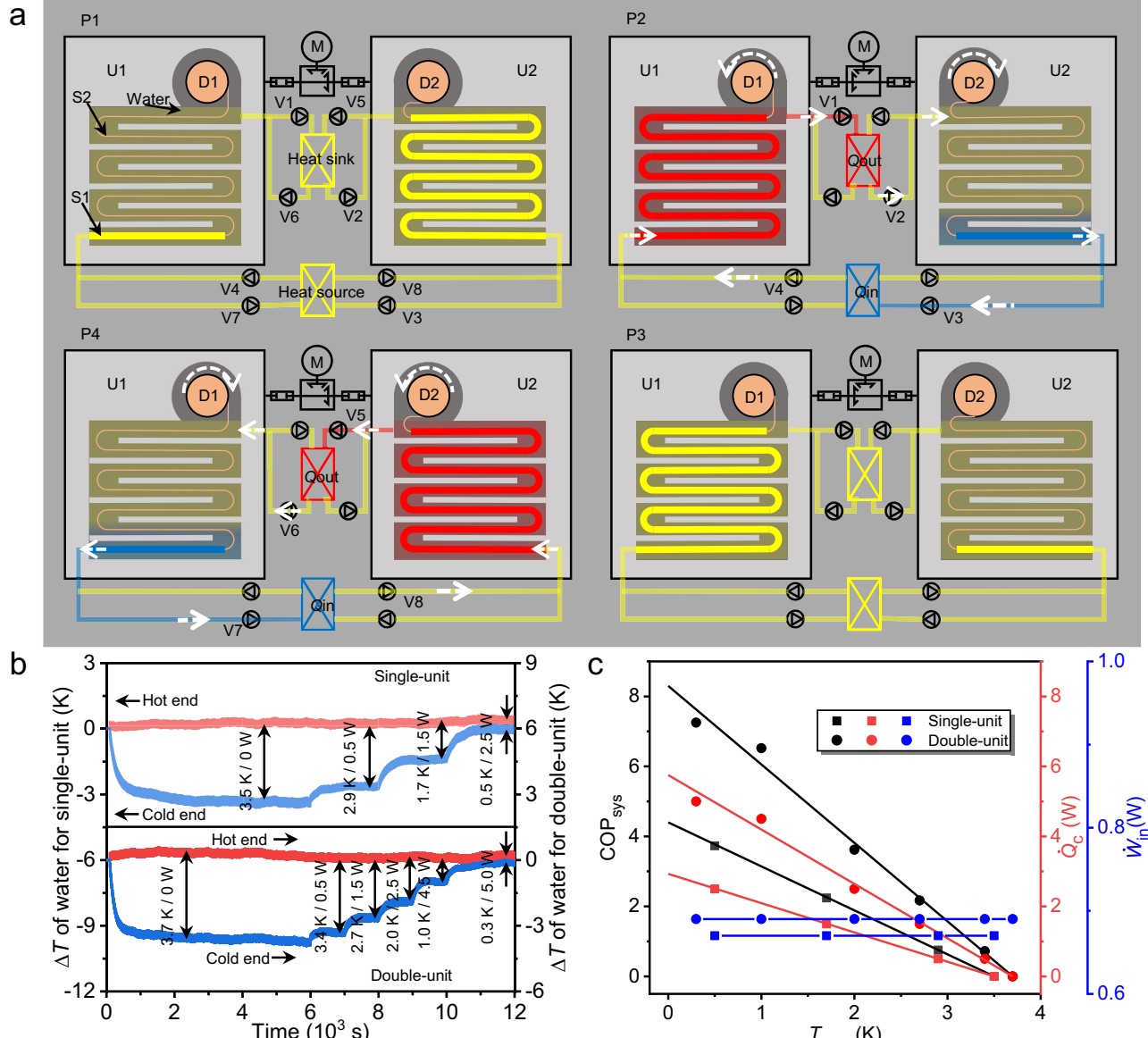

**Fig. 4 | The design and performance of the cooling devices. a** Ideal schematic diagram of a single cooling cycle for the designed double-unit cooling device under the strain level from 0% to 600%, where, M represents the servo motor, U1 and U2 represent unit 1 and 2, respectively, D1 and D2 represent the driving wheels of U1 and U2, respectively, S1 represents 1GNS/HH sample, S2 represents traction rope made of ultra-high molecular weight polyethylene fiber, V represents one-way pump, the white dashed arrows represent the directions of the water flow and directions of the rotation of driving wheels. U1 and U2 are coaxially connected to the right-angle conversion gearbox via two couplings. **b** $\Delta T$ of water versus time of the single-unit and the double-unit cooling devices under the flow rate of 1.2 mL min$^{-1}$ when the heat sink was maintained in an ambient environment and the heat source with different power values was kept in an insulated reservoir. **c** COP$_{sys}$, $\dot{Q}_c$, and $\dot{W}_{in}$ as a function of $T_{span}$ for the single-unit and double-unit cooling devices under a flow rate of 1.2 mL min$^{-1}$.

cooled by water during the process of P2 to P3. Moreover, the $T_{span}$ of the cooling devices decreases with the increase of the cooling power. The maximum cooling power (zero $T_{span}$) of the double-unit cooling device is approximately twice that of the single-unit cooling device.

The COP$_{sys}$ of these two cooling devices are calculated according to the Supplementary Equation (17). As illustrated in Fig.4c, the maximum COP$_{sys}$ for the single-unit and double-unit cooling devices are 4.4 and 8.3, respectively. The COP$_{sys}$ value of the double-unit cooling device is about 1.9 times that of the single-unit device, which is mainly due to the difference of input energy ($W_{in}$) of these two devices (details in Supplementary Discussion 4 and Supplementary Fig. 18). As illustrated in Supplementary Discussion 4 and Supplementary Fig. 19, when the refrigerant in the U1 unit of the double-unit device is stretched, the contraction work of the refrigerant in the coaxial connected U2 unit can be recovered and utilized. By contrast, the contraction

work in the single-unit device is wasted. During a single cooling cycle, the work required to drive the same mass of refrigerant material in the single-unit device is about 2.72 times that in the double-unit device. According to Supplementary Equation (18) and Supplementary Fig. 20, the work recovery efficiency of the double-unit device is calculated to as high as 56.3%.

In addition to the challenges discussed in this work, such as the $e$-CE response and the large-deformation characteristics of polymers, the applications of the polymer $e$-CE in solid-state cooling still relies on further development of other application-oriented cooling performances of polymers. These include the investigation of the fatigue life, thermal conductivity, thermal hysteresis, and compatibility with cooling devices. Supplementary Discussion 5 and Supplementary Tables 4, and 5 summarize and compare the cooling performances of a series of polymer and alloy materials, including the key parameters

such as external field strength, fatigue life, and thermal conductivity, which provide the relationship between the refrigerant parameters and cooling devices. For future research, it is necessary to integrate the reduction of the heat transfer period with the increase of refrigerant mass and the enhancement of fatigue life. This would lead to the overall performance of polymer-based cooling devices to meet application requirements.

## Discussion

In summary, the *e*-CEs enhanced by *s*-CE is attributed to the full utilization of the reversible conformational change in polymer chains assisted by the incorporated inorganic nanofillers. The construction of the co-existing shear and uniaxial deformation elements in a uniaxial tension polymer system has been demonstrated to improve the *e*-CE response by 20%–30% compared to that in pristine polymers. Moreover, the designed double-unit cooling system which adapted to the large-deformation characteristic of polymers not only possesses a high $COP_{sys}$ of 8.3 but also presents an impressive work recovery efficiency of 56.3%. In view of the flexibility of the polymer structure design and the nanofiller functionalization, we anticipate that the *s*-CE can be further optimized in the polymer composites. The composite strategy is anticipated to be widely utilized in the design of the mechanocaloric materials undergoing reversible conformational changes. Our work establishes a new approach to introducing additional CE entropy into polymer materials and demonstrates the potential of polymer composites in solid-state cooling applications.

## Methods

### Materials and preparation

SEBS thermoplastic elastomers were provided by KRATON Polymers Co., Ltd., TSRC Industries Co., Ltd., and Northwest Rubber & Plastics Research & Design Institute Co., Ltd., China. GNS were provided by Chengdu Chemicals Co., Ltd, Chinese Academy of Science. MCNT were provided by Chengdu Chemicals Co., Ltd, Chinese Academy of Science. CB were provided by Anhui Science Technology Co., Ltd., China. Methylbenzene was an analytical reagent obtained from Aladdin Bio-Chem Technology Co., Ltd. Specific product information see Supplementary Methods section. The above nanofillers were dispersed into methylbenzene under one hour's sonication to prepare the nanofiller solutions. Then, commercial SEBS were dissolved in methylbenzene by magnetic stirring for two hours at room temperature to prepare the matrix solutions with a solid content of 15 wt.%. After the matrix solutions were stirred to clear, the nanofiller solutions were proportionally added to the matrix solution and stirred for three days. The as-prepared slurries were transferred to glass culture dishes and dried at 60 °C for two days in a vacuum oven to remove the trace methylbenzene. Finally, the nanofillers/SEBS composites sample sheets (0.5–1 mm thickness) were obtained. Details for determining solvent residue in the sample sheets see Supplementary Methods. Test specimens for 2D WAXD, DMA and mechanical characterization and e-CE measurement were die cut from the same molded composite sample sheet. Film samples with a thickness of about 10 μm were prepared by solvent-casting and used for the AFM test. Details of the preparation conditions see Supplementary Methods. All samples were kept at the required ambient temperature for ten minutes before testing. The fabrication, denotation, and labeling of nanofillers/SEBS composites and the mass fraction of nanofillers are shown in Supplementary Table 1.

### The measuring device and the test methods of the *s*-CE enhanced *e*-CE $\Delta T_{adi}$

The $\Delta T_{adi}$ were directly measured by using a custom instrument, including a stretching device and temperature measuring device.

Details of the instrument see Supplementary Methods. The $\Delta T_{adi}$ was induced in an open indoor environment, heat sink and heat source involved in the heat exchange process were ambient air in this paper. Direct measurement of $\Delta T_{adi}$ was performed by measuring the in-suit surface temperature variation of the deforming test specimen at a strain rate (15 s$^{-1}$) higher than the adiabatic strain rate. Details of the method for determining the adiabatic strain rate see Supplementary Methods. The surface temperature of the deformed test specimen was measured by an on-line infrared thermometer (ABSD-01A, Aobosaide Automation Technology Co., LTD; BRW600-406, Hunan Firstrate Sencer Co., Ltd) and visualized by an infrared thermal imager (T890-2, Testo SE & Co. KGaA). The on-line infrared thermometer and the infrared thermal imager were fixed 10 cm above the plane of the test sample. The infrared emissivity of the infrared thermometer and the infrared thermal image was adjusted to that of the test sample (0.91), which was controlled by the built-in software of the thermal imager. The infrared emissivity of the test samples was determined by using a dual-band emissivity meter (IR-2, Shanghai Chengbo Photoelectric Technology Co., Ltd.) in 8–14 μm bands at room-temperature. Temperature information detected by an infrared thermometer was recorded at a frame rate of 9 s$^{-1}$, and it was averaged on a flat area of ~0.5 cm$^2$.

### Materials characterization

The molecular properties of these SEBS were assessed by gel permeation chromatography (GPC, PL-GPC50, Agilent technologies, CA, USA). The GPC measurements were carried out at 40 °C using THF as the eluent at a flow rate of 1 mL min$^{-1}$. The chemical structures of SEBS samples were analyzed by $^1$H-nuclear magnetic resonance ($^1$H-NMR) spectroscopy (Bruker, 400 MHz). TGA was performed with a heating rate of 10 K min$^{-1}$ in the temperature range of 298–1273 K under the nitrogen atmosphere. TEM images were obtained on FEI Tecnai G2 F30 with an accelerating voltage of 200 kV. The samples were ultra-microtomed at −130 °C to a section with a thickness of about 70 nm. The structure evolution of SEBS samples was in situ monitored by two-dimensional wide-angle X-ray diffraction (2D WAXD), which was performed using Genix 3D X beamline with wavelength λ = 1.54 Å at Xeuss 2.0 (Xenocs). The sample-to-detector distances for WAXD were set to be 151.7 mm. The value of the scattering wave vector magnitude is given by $q = \frac{4\pi \sin\theta}{\lambda}$ where $2\theta$ is the scattering angle. The 2D WAXS measurements were taken immediately after the sample reached the predetermined strain. The exposure duration was set as 120 s for each picture. Mechanical tests were run in an electromechanical universal testing machine (E44.104, MTS systems Co., LTD) equipped with a BSA-XS-50kg force transducer and a CEC1200 temperature testing chamber. Dynamic mechanical properties of SEBS samples were carried out on a dynamic mechanical analyzer (DMA, PE-DMA8000) at 1 Hz with a heating rate of 2 K min$^{-1}$ in the temperature range of 173–443 K. The specific heat capacity ($c_p$) was confirmed by the standard direct heat capacity measurement by using a differential scanning calorimetry (DSC, TA-DSC2500) instrument with a heating rate of 10 K min$^{-1}$ in the temperature range of 203–443 K. Before testing the specific heat capacity, sapphire standard sample was used as the reference material for the Tzero and Direct heat capacity calibration of TA-DSC2500. Both calibration processes were performed with a heating rate of 10 K min$^{-1}$ in the temperature range of 183–573 K. As the samples we used in this work did not involve a phase change such as strain-induced crystallization during the deformation process, the strain-dependence of $c_p$ was ignored. All composite samples were placed in liquid nitrogen overnight, and the cross-sections of those after brittle fracture were used to investigate the morphology and dispersion of nanofillers by field-emission scanning electron microscope (FE-SEM, Hitachi SU8010). Atomic force microscopy (AFM, Bruker Dimension icon) was employed for imaging the morphology and phase of the composite samples. All experiments were carried out with the same AFM probe

under ambient conditions (temperature of 25 °C, relative humidity of 25%). The morphological analyses and phase images were performed under ambient conditions using tapping mode. All quantitative measurements were carried out using a standard probe (0.01–0.025 Ohm-cm Antimony (n) doped Si, RTESP-150, Bruker) with a cantilever of 125 μm, resonant frequency of 150 kHz and spring constant $k$ of 6 N m$^{-1}$. Thermal conductivity was performed by the hot disk thermal analyzer (TPS2500S, Hot Disk Instrument) by using the transient plate heat source method (ISO22007-2).

## Model construction and simulation

Model construction and simulation were performed by the Materials Studio Package and the Martini forcefield. Details of the model building method see Supplementary Methods. Periodic microstructures of pure SEBS and 1GNS/HH were constructed by using cubic boxes with edges up to 26 nm followed by geometry optimization with a convergence threshold for the specified maximum energy change of $10^{-3}$ kcal mol$^{-1}$ and a convergence threshold for the specified maximum force of 0.5 kcal mol$^{-1}$ Å$^{-1}$. After that, a Mesocite Dynamics was employed to make these microstructures from a steady state to a stretched state. Detailed in Supplementary Methods. The whole simulation time was 4000 ps, resulting in a strain of 6 for both boxes. The coordinates of the first and last two beads of each molecular chain were derived from the result file, and the linear distance between the two coordinates was calculated as the end-to-end distance.

## Data availability

All data that support the findings of this study are included in the main text and Supplementary Information. Source data are provided with this paper.

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

## Acknowledgements

C.X. acknowledged the supported by the National Natural Science Foundation of China (No. 52273254, 51673154 and 51503159). Q.W. acknowledged the support of ACS PRF (66044-ND7). 2D WAXS was performed at School of Materials Science and Engineering of Hubei University.

## Author contributions

C.X. and Q.W. jointly supervised and directed this project. S.Z. conceived the idea, designed the device, and planned the project. S.Z., Y.F., and X.N. performed the measurements and the theoretical calculation. Y.W., Y.Z., C.L., J.Y., W.X., and Y.S provided experimental support. S.Q. discussed the result and provided suggestions on the device. S. Z. and Y. F. analyzed all data and wrote the first draft with discussion and input from all coauthors. C.X., Q.W., S.Q., Q.Y., and Q.L. provided insightful suggestions and revisions of the manuscript.

## Competing interests

The authors declare no competing interests.
