## [Peer Review File · Nature Communications]

Shearo-caloric effect enhances elastocaloric responses in polymer composites for solid-state coolingREVIEWER COMMENTS

Reviewer #1 (Remarks to the Author):

Dear authors,

thank you for submitting the summary of your research and developments in an emerging and relevant technology field.

The submitted article introduces a newly developed polymer composite for use in elastocaloric solid-state cooling applications.

The most noteworthy result of the broad study presented is the development of the specific polymer composite, which by design increases the elastocaloric effect under uniaxial stress. This approach opens up new ways of implementing polymer-based solid-state cooling or heating systems, reducing possible machine architecture and design in comparison to multi-axial stress configurations.

Solid-state cooling and heating plays a significant role in the current search for more environmental friendly and more energy-efficient alternatives to vapor-compression based cooling systems and heat pumps. In the field of polymer-based elastocaloric approaches, this novel material opens up new routes for possible applications in this field.

The work presented has been conducted extensively on material science and development side, includes experimental validation data accompanied by according modeling work, and first implementation on system level as exemplary efficient cooling system architecture.

The scientific methodology used is sound, including the supplemental information the submission has a good level of detail, the paper is easy to read, figures are of good quality.

I have several questions and recommendations, which in my opinion would increase the paper's quality:

General recommendation:

- You are listing the current challenges of polymers in solid-state cooling (high strains, latent heats, ΔT_s , ...), which are mostly addressed throughout the paper. I would suggest additional discussions on the challenges regarding life cycle numbers (what are expected statistical numbers with your developed material samples and what could create a promising outlook?). Another challenge for polymer structures is the low thermal conductivity. For efficient cooling systems, temperature exchange from the solid polymer to air or any fluid medium is crucial. A deeper discussion (and comparison to SMA materials, thin structures) would help to classify the current states of polymers vs. SMA based systems and thus help identification of possible application fields for each. Lastly, commentation on the new material's thermal hysteresis would be helpful in general with the system efficiency discussion mentioned above (quantification of efficiency losses on system level and possible improvements).

More specific suggestions:

- The exploitable latent heats in the materials are always quantified by the isothermal entropy change. For better direct comparison not only with competing polymer-based systems but with the currently most popular SMA-based approaches, it would be very helpful to also give a quantity on Enthalpie (dH in J/g) for relevant ambient temperatures, and also volumetric quantities for these latent heats. On system level, which defines construction space and weight, both quantities (volume and mass) defining possible cooling energy and power output are necessary. This could be included like in supplemental Fig. s3 in comparison to the state-of-research of other materials.

- Fig. 4b is a little confusing since it is not directly clear which graph is related to the single and

the double-unit.

- In the context of the work recovery on system level, it should be noted that work recovery through a range of architectures has been implemented in most recently realized elastocaloric system demonstrators and can be seen as general state-of-research on system level. For this discussion, a more complete literature review and references on realized hardware system demonstrators would be helpful, since there is only a few groups world wide, which have been able to present said realized hardware.

In the same context, the differentiation could be made by referencing more complex multi-axial approaches in the field of polymers (e.g. work of Kaltenbrunner, ...)

- Lastly, (not a must) it could be helpful in the title to mention the fact of polymer-based materials

Thank you for taking into account my suggestions,

best regards.

Reviewer #2 (Remarks to the Author):

The authors report how the inclusion of "shearo-caloric" elements into a polystyrene-b-poly(ethylene-co-butylene)-b-polystyrene polymer compound can effectively enhance the aggregate's elastocaloric performance. This is achieved by integrating a small mass of carbon nanofillers, which significantly enhances the effective elastic chains through shearing interlaminar molecular chains. This unlocks additional degrees of freedom that can contribute to the elastocaloric effect, translating to an adiabatic temperature change of -18.0 K and an isothermal entropy change of 187.4 J/kg·K. These values surpass previously reported polymers, including their previous work [Zhang, S., et al. "Solid-state cooling by elastocaloric polymer with uniform chain-lengths, Nat. Commun. 13 (2022) 1–7.]. They further develop a large-deformation cooling system with a work recovery efficiency of 90.7%.

Overall, the manuscript is scientifically sound, and results are well supported by various experimental techniques, including infrared thermometry, TEM, AFM, WAXS, and DSC. Characterisation of both the developed materials and the cooling device is quite thorough.

The inclusion of elements inducing shear onto the interlaminar molecular chains is an original strategy for improving elastocaloric properties in polymers. This strategy has the potential to open up new research opportunities for materials calorific optimisation. Moreover, presenting a GNS-HH-based cooling device with the interesting property of work recovery positively reinforces the goal of the study.

However, the manuscript requires a revision to address the following points:

- The manuscript requires linguistic improvement. It contains typographical and grammatical errors and, at times, lacks readability.

- Some figures should be more self-explanatory, and their captions should contain more specific information on the data shown on each panel. When necessary, indicate the main details of the experimental technique and all included labelling. For instance, indicate the strain at which data shown in Figure 1a has been collected.

- Please remove "(a.u.)" as the units of COP_{sys} in figure 4c.

- Regarding the specific heat capacity, the authors state, "The specific heat capacity (cp) was confirmed by differential scanning calorimetry (DSC, TA-DSC2500) measurement with a heating rate of 10 K min⁻¹ in the temperature range of 203–443 K." Is c_p obtained from a standard or modulated DSC measurement? How is it derived, considering that standard DSC alone is

inaccurate for "c_p" characterisation?

- Timescales for heat transfer are rather long, which is a frequent problem in caloric cooling. This hinders operation frequency, which, in turn, constrains the device's cooling power. Figure S13, for instance, displays information on the water flow rate. More details should be given on the selection of the 0.167 Hz operation frequency and the performance on other frequencies.

Reviewer #3 (Remarks to the Author):

This study focuses on the elastocaloric effect of SEBS/graphene composites. It covers the development and characterisation of the materials as well as their integration into cooling systems. This work is interesting and could be of interest to the solid-state cooling community for the following reasons: 1) it highlights a significant elastocaloric effect; 2) until now, elastocaloric polymer composites have not been studied intensively; 3) the recovery capacity of the system has also been tested. However, the reviewer identified several shortcomings in the manuscript that should be corrected before reconsidering its publication:

1) The title could mention that it concerns polymer composites.

2) The introduction to thermoplastic elastomers and composites is too short. Specifically, no work on the impact of deformation and the effect of carbon fillers on the structure and on the mechanical response of SEBS (or more generally on thermoplastic elastomers) is cited. However, there are a few works such as :

_Dechnarong ,Macromolecules 2020, 53, 20, 8901-8909

_Haonan Liu et al 2020 Jpn. J. Appl. Phys. 59 SN1013

_Yongjin Li Macromolecules 2009, 42, 7, 2587-2593

_Enrique-jimenez, European Polymer Journal Volume 97, December 2017, Pages 1-13

In addition, elastocaloric composites should also be clearly mentioned in the introduction. The following references can be added and commented:

_Nicolas Candau et al., Vol. 16, No.12, Pages 1331-1347, 2022

_Seok Bin Hong et al 2019 Funct. Compos. Struct. 1 015004

Finally, the bibliography concerning polymeric elastocaloric systems is not clearly mentioned.

3) Regarding material processing. It is written that the composites were agitated for 3 days. Did the authors ensure that there was no sedimentation of the particles?

4) Regarding the processing of the materials (in the additional information file). It is mentioned that the composites have a thickness that can vary between: 0.5 and 1 mm. ("And finally, sample sheets of nanofiller/SEBS composites (0.5-1 mm thick) were obtained").

This sentence raises two questions:

_For the solvent casting process, it is difficult to remove all the solvent when the thickness is greater than around 100-200 micrometers. Have the authors carried out a TGA (thermal gravimetric analysis) to check that no solvent is present inside the composites at the end of the process? If any residual solvent remains, what is the weight fraction of solvent for each formulation?

The second concern is that adiabatic conditions depend on the strain rate, thickness and thermal conductivity. The adiabatic deformation rate of 5 s⁻¹ mentioned in the supporting information concerns which thickness and which formulation? (thermal conductivity can be affected by the presence of carbon fillers (man, F.C.. J Mater Sci 58, 11029-11043 (2023) and the convective heat transfer constant is related to thickness (Mott et al., PolymerVolume 105, 22 November 2016, Pages 227-233) which varies with the formulation in this work). These points are important because the filler optimum determined via DT{adiab} measurements could be influenced by the way

it is measured. (If the measurement is not adiabatic, the temperature variations cannot be compared).

5) Regarding the material processing for AFM, it is not the same as that used for the rest of the study. However, it is known that a skin effect can occur when using the solvent casting method (K. Wongtimnoi et al. ,Volume136, Issue3, January 15, 2019) and my concern is that the AFM observation may not be representative of the bulk structure. I would recommend that the authors use the samples obtained with the conventional process (leading to a thickness of about 500µm) to prepare the surface of the sample for AFM by cryo ultra-microtomy.

6) In the results and discussion section, P5 lines 4-5, the authors should clearly mention: the molecular chain length, side group content and weight fraction of the PS/PEB so that the reader can easily relate it to their previous study. They should also provide references to the polymers supplied by KRATON and Northwest Rubber & Plastics Research & Design Institute Co, Ltd in the supplementary informations. They should comment on Figure S1 and indicate the molecular weight in supplementary section. For HH and LH, the molecular weight is not readable in Figure S1.

7) In the results and discussion section, p5 line 8, I don't understand what "ultimate strains" means (although this point seems important for comparing DTadi)? Is it just below the strain at break? Is it 600% for each sample (Figure 1.b)? Is it also possible to add adiabatic temperature values for different strains?

8) In the results and discussion section, for the sentence "This room temperature e-CE activity exceeded those of previously reported elastocaloric polymers (Fig. s3)" the authors should also refer to the work of Greibich et al. which claims a temperature variation of around 23K (Greibich, F et al. Nat Energy 6, 260-267 (2021).) and also the work of Run Wang et al (SCIENCE Oct 2019 Vol 366, Issue 6462 pp. 216-221) and perhaps adjust this sentence . They could also adapt the sentence in the abstract: "The consequential adiabatic temperature change of -18.0 K and isothermal entropy change of 187.4 J kg⁻¹ K⁻¹ exceed those of other elastocaloric polymers."

9) Figure. 2, the strain at which the characterisation was carried out is not mentioned. This point is really important especially for figure.2.k. It might also be interesting to see how it evolves for each formulation with strain since in situ SEBS study shows that it is not monotonous (Dechnarong ,Macromolecules 2020, 53, 20, 8901-8909). Besides, how long after deformation was the Xray measurement taken?(Was the stress stable?) For figures2.h) and 2.i), why the maximum of intensity is lower after stretching for 1GNS/HH (green in the colour scale) than for HH (yellow in the colour scale)? My concern is that the peak should be thinner and higher for 1GNS/HH (not only thinner).

10) Figure.2 , lower magnification images could be added in TEM and AFM. From AFM observation (Figure 2.d)), it appears that the composites are made up of huge aggregates and therefore the number of chains interacting with the fillers should not be high and perhaps not in agreement with Figure 2.a) which suggests that most chains interact with the fillers.

11) With regard to end-to-end determination, is it possible to use the Herman function directly in nanocomposites of thermoplastic block polymers as you did or is it only possible in homopolymers without fillers? Please mention other work using this function in block copolymer composites.

12) With regard to end-to-end determination and affine deformation:

_ how is Figure.2.m obtained? The authors should mention in the main text that it is extracted from the statistical study presented in the supplementary information (which is carried out under the assumption of affine deformation).

_Can the affine deformation assumption and entropy elasticity formula (equ s2 , S5 s6) be used for thermoplastic elastomers composites? Authors may cite papers to support this, because it looks like to be in disagreement with "Dechnarong ,Macromolecules 2020, 53, 20, 8901-8909" since it was shown that a non affine deformation occurs for SEBS polymers above a critical strain and also because plasticity is observed in figures.s7 and figure.s13.

13) In the paragraph "Structure evolution and enhancement mechanism of CE for GNS/HHs", no

references are cited to support and/or compare the conclusions, i.e. for example, references relating to the impact of stress on the stretching of macromolecules and on the interaction of GNS with PS blocks could be added.

14) Page 9 lines 14-18: "With the increase in GNS loading from 0 to 1 wt%, the maximum values of $\tan \delta$ corresponding to Tg-PEB progressively decreased, and those corresponding to Tg-PS progressively increased". In this case, why for 5 and 8% of GNS loading, the peak associated with PEB Tg is higher and that associated with PS Tg is lower than the unloaded HH matrix and why the rubbery plateau is much lower for 8 GNS/HH. Furthermore, the authors should probably refer to the alpha relaxation temperature rather than the glass transition temperature. Finally, they could perhaps use literature such as the work of S. Kuester et al / Composites Part B 84 (2016) 236-247 to compare and interpret their data.

15) Regarding Table S1 and Figure S4, the reviewer does not understand why so much data is presented for carbon black and carbon nanotube composites, since structure and temperature variation are only evaluated for "GNS/HH" composites in the main part of the paper.

16) In the Double-Unit cooling device with work capacity recovery section, the authors could compare the performances of their system with those of Greibich et al (Greibich, F et al. Nat Energy 6, 260-267 (2021)), Sebald et al (Sebald et al. Applied Thermal Engineering Volume 223, 25 March 2023, 120016) and Run Wang et al (SCIENCE Oct 2019 Vol 366, Issue 6462 pp. 216-221).

17) In the Double-Unit cooling device with work capacity recovery section, The massic cooling power (cooling power divided by mass of active materials) could also be presented to facilitate comparison between literature studies.

18) The recovery capacity is really interesting. This makes sense of the COP_{mat} in which the heat that can be exchanged is divided by the mechanical hysteresis assuming that the unloading work can be reused (Cui et al. Applied Physics Letters 101, 073904 (2012)). The authors could try to better explain the link between the mechanical hysteresis of the material (outside the system) and the input work which is required in the system. The ideal consumed energy and the real consumed energy are not clear for me, so please improve the quality of the explanation of equation 18. Besides, the authors could check the literature for other systems that prove the possibility of using recovery work and mention the associated literature if it exists, otherwise I recommend then to better highlight this part of their work if it is new.

Response to reviewers' comments

Shearo-caloric elements enhanced elastocaloric effects for solid-state cooling (NCOMMS-24-01724-T)

Dear Reviewers:

We thank you for your handling and suggestions about our manuscript. We appreciate you for providing positive feedback and a thorough evaluation of our work. We have fully revised our manuscript and addressed all the reviewers' comments by adding new data and analyses/discussions, which have greatly strengthened this work.

The major revisions and new analyses we have undertaken are summarized below and discussed in detail in the point-by-point responses. All changes in the manuscripts were highlighted with tracked changes in the revised text files.

We would appreciate your time and further consideration of our revised manuscript for publication in *Nature Communications*.

Best regards

Reviewer #1 (Remarks to the Author)

Dear authors,

Thank you for submitting the summary of your research and developments in an emerging and relevant technology field.

The submitted article introduces a newly developed polymer composite for use in elastocaloric solid-state cooling applications.

The most noteworthy result of the broad study presented is the development of the specific polymer composite, which by design increases the elastocaloric effect under uniaxial stress. This approach opens up new ways of implementing polymer-based solid-state cooling or heating systems, reducing possible machine architecture and design in comparison to multi-axial stress configurations.

Solid-state cooling and heating plays a significant role in the current search for more environmental friendly and more energy-efficient alternatives to vapor-compression based cooling systems and heat pumps. In the field of polymer-based elastocaloric approaches, this novel material opens up new routes for possible applications in this field.

The work presented has been conducted extensively on material science and development side, includes experimental validation data accompanied by according modeling work, and first implementation on system level as exemplary efficient cooling system architecture.

The scientific methodology used is sound, including the supplemental information the submission has a good level of detail, the paper is easy to read, figures are of good quality.

I have several questions and recommendations, which in my opinion would increase the paper's quality:

Response: We thank you so much for the acknowledging and comprehensive review of this study. We appreciate your comments about the life cycle numbers, thermal conductivity, thermal hysteresis, and specific suggestions that inspired us to comprehensively investigate the issues that still need to be addressed in the elastocaloric cooling application, which will be the topics for our subsequent studies.

General recommendation:

Question 1. *You are listing the current challenges of polymers in solid-state cooling (high strains, latent heats, deltaTs, ...), which are mostly addressed throughout the paper. I would suggest additional discussions on the challenges regarding life cycle numbers (what are expected statistical numbers with your developed material samples and what could create a promising outlook?).*

Response: Thank you for your in-depth comment. Fatigue life is one of the most important metrics for applications. The statistical numbers with these materials are about 10^3 cycles, and the long-term application of polymers has not been given much consideration when compared to the better-developed SMA materials. This work mainly delves into the persistent challenge of inadequate elastocaloric response encountered by polymers, alongside the designing of the cooling devices capable of accommodating large deformations. 10^3 cycles of loading and unloading are sufficient for us to obtain stable elastocaloric responses of the polymer and stable device performance.

To address this comment, we added application-oriented discussions, comparisons, and references on the life cycle numbers. The changes in the revised manuscript are as follows:

(1) Text

In addition to the challenges discussed in this work, such as the e-CE response and the large-deformation characteristics of polymers, the applications of the polymer e-CE in solid-state cooling still relies on further development of other application-oriented cooling performances of polymers. These include the investigation of the fatigue life, thermal conductivity, thermal hysteresis, and compatibility with cooling devices. Supplementary Discussion 5 and Supplementary Tables 4 and 5 summarize and compare the cooling performances of a series of polymer and alloy materials, including the key parameters such as external field strength, fatigue life, and thermal conductivity, which provide the relationship between the refrigerant parameters and cooling devices. For future research, it is necessary to integrate the reduction of the heat transfer period with the increase of refrigerant mass and the enhancement of fatigue

life. This would lead to the overall performance of polymer-based cooling devices to meet application requirements.

(Line 15 of Page 13 in the Main Manuscript)

(2) Text

Supplementary Discussion 5. Comparison of the application-oriented cooling performances and the cooling devices among polymer and SMA.

The *e*-CE activities and the large-deformation feature of polymers, which are discussed in the main text, are the most fundamental challenges for polymers to move toward solid-state cooling applications. In addition to the above performances, other cooling performances of different materials and associated device performances are summarized in Supplementary Table 4, which would help to classify the current work.

Commercial air conditioners, based on vapor-compression cooling technology, generally need to be recharged with additional refrigerant approximately every 5 years. This is mainly caused by the gradual leakage of gas refrigerant. For solid-state cooling applications, the frequency of the replacement of solid refrigerant is determined by the fatigue life of refrigerant materials and the operation frequency of the cooling device. With 0.167 Hz operation for a usage modality of 12 hours per day and 180 days per year,³¹ 6.5×10^6 cycles correspond to 5 years, which could meet the service life of commercial applications. NiTi alloy almost meets this requirement as shown in Supplementary Table 4.^{30,31} For the large-deformation refrigerants, only the cycling fatigue life of NR for *e*-CE cooling has been systematically studied.³² Without considering the weather resistance, the result shows a high fatigue life of up to 1.7×10^5 cycles for NR samples with little degradation of *e*-CE properties, constituting an important demonstration of cooling application. The fatigue life of large-deformation refrigerant has not been determined further, which is due to the focus of current research being on the design and testing of the cooling device.^{19,20,26} For 1GNS/HH in this work, $\sim 10^3$ cycles of fatigue life are achieved from testing of the cooling device. At the materials level, Supplementary Fig. 21 shows little degradations of the mechanical strength and *e*-CE activity of 1GNS/HH after the first 10 cycles. Moreover, the fatigue life of SEBS can be improved to 8.8×10^4 cycles through

further modifications, which makes it possible to optimize the material composition to meet practical requirements.³³ The mechanism of fatigue fracture of SEBS and the corresponding performance improvements still require further research.

(Line 1 of Page 20 in the Supplementary Information)

(3) Table

Supplementary Table 4. Performance comparison of e -CE cooling systems and materials.

Sample	Status	Refrigerant		Cycles	Medium	System					Ref.
		$\Delta\sigma$ (MPa)	ε (100%)			COP_{sys}	T_{span} (K)	f (Hz)	\dot{Q}_c (W)	SCP (W g ⁻¹)	
NiTi	Membrane Sheet	450	0.034	2×10^3	Water	3.5	15.3	0.25	4.64	0.8	27
TiNiFe	Membrane Foil	500	0.055	–	Cu metal	3.2	13	4	7.9	7.7	28
NiTi	Membrane Sheet	–	0.043	6×10^3	Water	–	19.9	–	–	–	29
NiTi	Bulk tube	1234	–	10^7	Water @	–	5.6	1.2	7.9	6.27	30
NiTi	Bulk tube	700	0.035	7.5×10^4	Water	6.85	22.5	0.071	260	0.3	31
NR	Bulk fiber	–	0-1 #	750	Water	–	0.7 ##	–	–	–	19
NR	Membrane balloon	*	**	10^3	Al metal	–	7.9 ***	1.1	0.75	20.9	20
NR	Bulk tube	1.5	3.5-5.5	3×10^4	Water	6	8.3	0.1	1.5	0.14	26
1GNS/HH	Bulk film	4	1-5	10^3	Water	8.3	3.7	0.167	5.0	5.47	This work

T_{span} is collected near zero \dot{Q}_c . \dot{Q}_c and COP_{sys} are collected near zero T_{span} .

@ The heat is transferred by evaporation and condensation of water.

Accompanying by isometrical twisting and untwisting at 15 turns/s. ## Assume that the temperature drop of the water is the T_{span} under zero \dot{Q}_c .

* 8.5 kPa of gas pressure. ** Balloon volume change from 115 to 200 cm³. *** Temperature span between the hot and cold sides of a device.

(Page 49 in the Supplementary Information)

(4) Figure

(Page 43 in the Supplementary Information)

(5) Reference

19. Wang, R. et al. Torsional refrigeration by twisted, coiled, and supercoiled fibers. *Science* **366**, 216-221 (2019).
20. Greibich, F. et al. Elastocaloric heat pump with specific cooling power of 20.9 W g⁻¹ exploiting snap-through instability and strain-induced crystallization. *Nat. Energ.* **6**, 260-267 (2021).
26. Sebald, G., et al. High-performance polymer-based regenerative elastocaloric cooler. *Appl. Therm. Eng.* **223**, 120016 (2023).
30. Bachmann, N. et al. Long-term stable compressive elastocaloric cooling system with latent heat transfer. *Commun. Phys.* **4**, 194 (2021).
31. Qian, S. et al. High-performance multimode elastocaloric cooling system. *Science* **380**, 722-727 (2023).
32. Sebald, G., Xie, Z. & Guyomar, D. Fatigue effect of elastocaloric properties in natural rubber. *Phil. Trans. R. Soc. A* **374**, 20150302 (2016).
33. Maji, P. et al. Strategic fabrication of SEBS composite with high strength and stretchability

via incorporation of polymer-grafted cellulose nanofibers for biomedical applications. *Cellulose* **30**, 9465-9484 (2023).

(Page 52-53 in the Supplementary Information)

Question 2. *Another challenge for polymer structures is the low thermal conductivity. For efficient cooling systems, temperature exchange from the solid polymer to air or any fluid medium is crucial. A deeper discussion (and comparison to SMA materials, thin structures) would help to classify the current states of polymers vs. SMA based systems and thus help identification of possible application fields for each.*

Response: Thank you for your comment. We strongly agree with you that the thermal conductivity, heat transfer medium, and sample size are important for cooling systems. A comprehensive comparison of the performance metric related to practical applications could help to have better understanding of elastocaloric polymers.

To address this comment, we added more discussions, comparisons, and references to the revised manuscript and Supporting Information. The changes in the revised manuscript are as follows:

(1) Text

Another challenge for polymers is heat transfer. As illustrated in Supplementary Table 5, GNS/HHs possess the maximum thermal conductivity of $0.4853 \text{ W m}^{-1} \text{ K}^{-1}$, which is two orders of magnitude smaller than that of alloy materials. This means that the heat transfer rate of polymer is lower than that of alloy under the conditions of the same shape of refrigerant and the same heat transfer medium. As illustrated in Supplementary Table 4, when the heat transfer medium is water and the shape of the refrigerant is bulk tube,^{26,30} the low thermal conductivity further hinders the operation frequency and specific cooling power of the associated polymer-based cooling device in order to ensure adequate heat exchange. In this case, the polymer-based cooling device could still maintain a higher temperature span. In this work, water flow is used as a heat transfer medium to extract the cooling energy generated from bulk refrigerant. As

another major advantage of this configuration being the scalability of the refrigerant, it is possible to increase the mass of refrigerant, thereby increasing the cooling power without significantly altering the device structure. Another configuration of the cooling device uses a thinner refrigerant to transfer heat through solid-solid contact.^{20,28} This configuration can significantly reduce the time required for heat transfer because of its high specific surface area, thereby achieving higher operating frequency and specific cooling power. This makes the operation frequency of polymer-based cooling devices comparable to that of alloy-based devices while achieving a specific cooling power of up to 20.9 W g⁻¹. However, the cooling power of the polymer-based cooling device is much lower under this configuration. This is because specific cooling power is related to both operation frequency and sample shape (specific surface area). As the value of specific cooling power does not take into consideration of the mass or volume factors associated with the change of specific surface area, any changes in heat transfer caused by the mass/volume factors or scalability have not been considered. In summary, to improve the overall performance of polymer-based cooling devices, it is necessary to further combine the reduction of the heat transfer period with the increase of refrigerant mass. (Line 5 of Page 21 in the Supplementary Information)

(2) Methods

Thermal conductivity measurement was performed by the hot disk thermal analyzer (TPS2500S, Hot Disk Instrument) by using the transient plate heat source method (ISO22007-2).

(Line 5 of Page 7 in the Supplementary Information)

(3) Table

Supplementary Table 4. Performance comparison of *e*-CE cooling systems and materials.

Sample	Status	Refrigerant			System						Ref.
		$\Delta\sigma$ (MPa)	ε (100%)	Cycles	Medium	COP_{sys}	T_{span} (K)	f (Hz)	\dot{Q}_c (W)	SCP (W g ⁻¹)	
NiTi	Membrane Sheet	450	0.034	2×10^3	Water	3.5	15.3	0.25	4.64	0.8	27
TiNiFe	Membrane Foil	500	0.055	–	Cu metal	3.2	13	4	7.9	7.7	28
NiTi	Membrane Sheet	–	0.043	6×10^3	Water	–	19.9	–	–	–	29
NiTi	Bulk tube	1234	–	10^7	Water @	–	5.6	1.2	7.9	6.27	30
NiTi	Bulk tube	700	0.035	7.5×10^4	Water	6.85	22.5	0.071	260	0.3	31
NR	Bulk fiber	–	0-1 #	750	Water	–	0.7 ##	–	–	–	19
NR	Membrane balloon	*	**	10^3	Al metal	–	7.9 ***	1.1	0.75	20.9	20
NR	Bulk tube	1.5	3.5-5.5	3×10^4	Water	6	8.3	0.1	1.5	0.14	26
1GNS/ HH	Bulk film	4	1-5	10^3	Water	8.3	3.7	0.167	5.0	5.47	This work

T_{span} is collected near zero \dot{Q}_c . \dot{Q}_c and COP_{sys} are collected near zero T_{span} .

@ The heat is transferred by evaporation and condensation of water.

Accompanying by isometrical twisting and untwisting at 15 turns/s. ## Assume that the temperature drop of the water is the T_{span} under zero \dot{Q}_c .

* 8.5 kPa of gas pressure. ** Balloon volume change from 115 to 200 cm³. *** Temperature span between the hot and cold sides of a device.

(Page 49 in the Supplementary Information)

(4) Table

Supplementary Table 5. Thermal conductivity of GNS/HHs at 298 K.

Sample	Thermal conductivity (W m ⁻¹ K ⁻¹)
HH	0.1673±0.0001
1GNS/HH	0.1960±0.0002
8GNS/HH	0.4853±0.0003

(Page 50 in the Supplementary Information)

(5) Reference

20. Greibich, F. et al. Elastocaloric heat pump with specific cooling power of 20.9 W g⁻¹ exploiting snap-through instability and strain-induced crystallization. *Nat. Energ.* **6**, 260-267 (2021).
26. Sebald, G., et al. High-performance polymer-based regenerative elastocaloric cooler. *Appl. Therm. Eng.* **223**, 120016 (2023).
27. Tušek, J. et al. A regenerative elastocaloric heat pump. *Nat. Energ.* **1**, 16134 (2016).
28. Bruederlin, F., Ossmer, H., Wendler, F., Miyazaki, S. & Kohl, M. SMA foil-based elastocaloric cooling: from material behavior to device engineering. *J. Phys. D Appl. Phys.* **50**, 424003 (2017).
29. Engelbrecht, K. et al. A regenerative elastocaloric device: experimental results. *J. Phys. D Appl. Phys.* **50**, 424006 (2017).
30. Bachmann, N. et al. Long-term stable compressive elastocaloric cooling system with latent heat transfer. *Commun. Phys.* **4**, 194 (2021).

(Page 52-53 in the Supplementary Information)

Question 3. Lastly, commentation on the new material's thermal hysteresis would be helpful in general with the system efficiency discussion mentioned above (quantification of efficiency losses on system level and possible improvements).

Response: Thank you for your in-depth recommendation. We agree with you that the thermal hysteresis of material is critical for the system. *To address this comment*, we have made following changes:

(1)

During the stable cycles (Fig. 1b), the temperature changes between P1-P2 (ΔT_H) is higher than that of P3-P4 ($|\Delta T_{adi}|$). This is mainly due to the fact that the polymer in this work only undergoes conformational changes (without first-order transition) during deformation. Therefore, the differences between ΔT_H and $|\Delta T_{adi}|$ during the heating and cooling processes are mainly caused by the internal friction within the molecular chains. The irreversible heat generated by internal friction could increase the temperature change caused by conformational arrangement during the heating process and decrease the conformation-induced temperature change during the cooling process. The irreversible heat generated by internal friction could be considered as the thermal hysteresis (ΔT_{hys}) of this material, and calculated as $\Delta T_{hys} = \frac{1}{2} (\Delta T_H - |\Delta T_{adi}|) = 3.6$ K. The ΔT_{hys} represents the irreversible portion of material during the *e*-CE cycle. Therefore, reducing ΔT_{hys} helps improve system efficiency, which can be achieved by adding plasticizers or other methods to reduce internal friction.

(Line 21 of Page 12 in the Supplementary Information)

(2)

On the other hand, W_i can be considered from the material level. As illustrated in Supplementary Fig. 20, when assuming that the unloading work of the refrigerant could be completely reused, the integrated area of the mechanical hysteresis (red-shaded area) is the energy consumed per unit volume of the refrigerant for a complete stretch-recovery cycle. This consumption represents the loss of work at the material level, which is caused by the material properties (internal friction).

(Line 18 of Page 18 in the Supplementary Information)

(3)

Supplementary Fig. 20. Stress-strain curves of 1GNS/HH during stable cycling. The black arrows point to the direction of strain development. The red shadow represents the integral area of the mechanical hysteresis. The blue shadow represents the integral area of the recovery curve.

(Page 42 in the Supplementary Information)

More specific suggestions:

Question 4. *The exploitable latent heats in the materials are always quantified by the isothermal entropy change. For better direct comparison not only with competing polymer-based systems but with the currently most popular SMA-based approaches, it would be very helpful to also give a quantity on Enthalpie (dH in J/g) for relevant ambient temperatures, and also volumetric quantities for these latent heats. On system level, which defines construction space and weight, both quantities (volume and mass) defining possible cooling energy and power output are necessary.*

This could be included like in supplemental Fig. s3 in comparison to the state-of-research of other materials.

Response: Thank you for your comment. We strongly agree with you that both these two quantities on isothermal entropy change are necessary for better comparison especially when defining construction space and weight. *To address this comment*, the volumetric quantities of isothermal entropy changes and isothermal enthalpy changes have been added in Supplementary Fig. 6 (Fig. s3 in the first draft). We also added two new citations for the comparison and made a correction to the data in Ref. 25.

The change in the revised manuscript is as follows:

(1)

(Page 28 in the Supplementary Information)

(2)

19. Wang, R. et al. Torsional refrigeration by twisted, coiled, and supercoiled fibers. *Science* **366**, 216-221 (2019).

20. Greibich, F. et al. Elastocaloric heat pump with specific cooling power of 20.9 W g⁻¹ exploiting snap-through instability and strain-induced crystallization. *Nat. Energ.* **6**, 260-267 (2021).

(Page 52 in the Supplementary Information)

Question 5. Fig. 4b is a little confusing since it is not directly clear which graph is related to the single and the double-unit.

Response: To address this comment, we added an auxiliary line for segmentation in the figure and the corresponding annotations.

The change in the revised manuscript is as follow:

Figure 4. The design and overall performance of the cooling devices. **b** ΔT of water versus time of the single-unit and the double-unit cooling devices under the flow rate of 1.2 mL min⁻¹ when the heat sink was maintained in an ambient environment and the heat source with different power values was maintained in an insulated reservoir.

Question 6. *In the context of the work recovery on system level, it should be noted that work recovery through a range of architectures has been implemented in most recently realized elastocaloric system demonstrators and can be seen as general state-of-research on system level. For this discussion, a more complete literature review and references on realized hardware system demonstrators would be helpful, since there is only a few groups world wide, which have been able to present said realized hardware. In the same context, the differentiation could be made by referencing more complex multi-axial approaches in the field of polymers (e.g. work of Kaltenbrunnner, ...)*

Response: Thank you for your in-depth comments. We agree with you that the work recovery performance of a CE device could be seen as a general state of research on the system level. This performance has been achieved in electrocaloric systems (such as E. Defay's work) and alloy materials (such as Ref. 14 and Kirsch, S. M.'s work). We added the related citations in the revised text. However, the implementation of the energy recovery performance remains challenging in the current large-deformation system. To the best of our knowledge, the work recovery performance has not been reported so far in the large deformation refrigerants or polymer systems.

The work of Kaltenbrunnner's group showcased the unique characteristics of polymer elastomers. This is a very interesting work, as the control of strain by blowing balloon is a unique form of polymer deformation. The device in Kaltenbrunnner's work (Fig. 4 in their work) used an incompressible dielectric elastomer actuator (DEA) as the electrical driving system (CE inactivity unit) and had no work recovery performance. The incompressible DEA is an electro-deformation capacitor, and it can only change its thickness as a result of the Maxwell stress. Therefore, during the CE cycle, DEA only undergoes charge and discharge and changes its thickness. The surface area change of DEA caused by its thickness change is not affected by gas pressure. Therefore, the recyclable work of the elastocaloric active unit (elastomer membrane) cannot be collected by DEA directly under the current conditions.

If their device needs to achieve work recovery performance, at least additional external devices need to be designed to collect and reuse the electrical energy generated by the DEA

discharge process. However, it is difficult to design an external recovery device for the electric energy discharged by the DEA, because it involves many mutually restrictive issues such as charging rate, capacity, and adaptability of the main device.

In addition to the work of Kaltenbrunner and E. Defay, Gael Sebald and coauthors have developed a series of regenerative elastocaloric cooling device, which greatly enriched the development of elastocaloric cooling devices.

To address this comment, we added more literature review and references on the hardware system demonstrators. The changes in the revised main text are as follows:

(1) Text

On the other hand, solid-state cooling critically depends on functional development of the cooling device that are tailored to **solid refrigerant materials and their responses to the external fields**.^{14-15, 32-37} **However**, the inherent large-deformation characteristics of polymers limit the development of their *e*-CE cooling devices **due to the demand for large driving strokes**. Only a few works have recently been reported on the large-deformation cooling polymer system.^{21, 22, 29, 38} For example, by harnessing snap-through instability in the soft capacitor to drive the expansion and contraction of an NR membrane balloon, a high operating frequency function was achieved for the *e*-CE cooling device.²¹ Another type of large-deformation cooling device utilizes the geometric design of NR tubes to endow the device with a thermal conductivity compensation function.^{22, 38} However, the energy recovery has not yet been realized in the large-deformation systems, which is crucial for further improving the system efficiency.³⁹

(Line 4 of Page 4 in the Main Manuscript)

(2) Reference

32. Defay, E., et al. Enhanced electrocaloric efficiency via energy recovery. *Nat. Commun.* **9**, 1827 (2018).
33. Li, J., et al. High cooling performance in a double-loop electrocaloric heat pump. *Science* **382**. 801-805 (2023).
34. Cui, H., et al. Flexible microfluidic electrocaloric cooling capillary tube with giant specific device cooling power density. *Joule* **6**, 258-268 (2022).

35. Ahčin, Ž., et al. High-performance cooling and heat pumping based on fatigue-resistant elastocaloric effect in compression. *Joule* **6**, 2338-2357 (2022).
36. Bruederlin, F., Ossmer, H., Wendler, F., Miyazaki, S. & Kohl, M. SMA foil-based elastocaloric cooling: from material behavior to device engineering. *J. Phys. D Appl. Phys.* **50**, 424003 (2017).
37. Kirsch, S. M. et al. NiTi-based elastocaloric cooling on the macroscale: From basic concepts to realization. *Energy Technol.* **6**, 1567–1587 (2018).
38. Sebald, G., et al. High-performance polymer-based regenerative elastocaloric cooler. *Appl. Therm. Eng.* **223**, 120016 (2023).
39. Qian, S. Thermodynamics of elastocaloric cooling and heat pump cycles. *Appl. Therm. Eng.* **219**, 119540 (2023).

(Page 17-18 in the Main Manuscript)

Question 7. *Lastly, (not a must) it could be helpful in the title to mention the fact of polymer-based materials*

*Thank you for taking into account my suggestions,
best regards.*

Response: Thank you for your kind suggestions. To address this comment, we have changed the title to "Shearo-caloric effect enhanced elastocaloric responses in polymer composites for solid-state cooling"

Thank you again for your comprehensive and in-depth recommendations and suggestions,

Reviewer #2 (Remarks to the Author)

The authors report how the inclusion of “shearo-caloric” elements into a polystyrene-b-poly(ethylene-co-butylene)-b-polystyrene polymer compound can effectively enhance the aggregate's elastocaloric performance. This is achieved by integrating a small mass of carbon nanofillers, which significantly enhances the effective elastic chains through shearing interlaminar molecular chains. This unlocks additional degrees of freedom that can contribute to the elastocaloric effect, translating to an adiabatic temperature change of -18.0 K and an isothermal entropy change of 187.4 J/kg·K. These values surpass previously reported polymers, including their previous work [Zhang, S., et al. "Solid-state cooling by elastocaloric polymer with uniform chain-lengths, Nat. Commun. 13 (2022) 1–7.]. They further develop a large-deformation cooling system with a work recovery efficiency of 90.7%.

Overall, the manuscript is scientifically sound, and results are well supported by various experimental techniques, including infrared thermometry, TEM, AFM, WAXS, and DSC. Characterisation of both the developed materials and the cooling device is quite thorough.

The inclusion of elements inducing shear onto the interlaminar molecular chains is an original strategy for improving elastocaloric properties in polymers. This strategy has the potential to open up new research opportunities for materials caloric optimisation. Moreover, presenting a GNS-HH-based cooling device with the interesting property of work recovery positively reinforces the goal of the study.

However, the manuscript requires a revision to address the following points:

Response: We thank you so much for the recommendation of this study. We appreciate your valuable comments on the operation frequency that inspired us to consider the situation of incomplete heat exchange, which will be the topics for our subsequent studies.

Question 1. *The manuscript requires linguistic improvement. It contains typographical and grammatical errors and, at times, lacks readability.*

Response: Thank you for your kind suggestions. We have thoroughly polished the manuscript and corrected the typos and errors.

Question 2. Some figures should be more self-explanatory, and their captions should contain more specific information on the data shown on each panel. When necessary, indicate the main details of the experimental technique and all included labelling. For instance, indicate the strain at which data shown in Figure 1a has been collected.

Response: Thank you for your kind suggestions. To address this comment, we added more details of experimental techniques in the figures to make the figures more self-explanatory.

The change in the revised manuscript is as follows:

(1)

(2)

(3)

(4)

Figure 2. The structure evolution and the enhancement mechanism of s-CE for GNS/HHs. j Magnification image for 1GNS/HH under P2 with enhanced contrast. The white arrows point to the (002) crystal plane of GNS.

(5)

Figure 2. The structure evolution and the enhancement mechanism of s-CE for GNS/HHs. m The distribution of the end-to-end distance of GNS/HHs before and after elongation. The x and z axes represent the centroid coordinates of the molecular chains, and the color scale represents the end-to-end distance.

(6)

Figure 4. The design and overall performance of the cooling devices. b ΔT of water versus time of the single-unit and the double-unit cooling devices under the flow rate of 1.2 mL min^{-1} when the heat sink was maintained in an ambient environment and the heat source with different power values was maintained in an insulated reservoir.

Question 3. Please remove “(a.u.)” as the units of COP_{sys} in figure 4c.

Response: Thank you for your comment. We removed "a.u." as the units of COP_{sys} . The change in the revised manuscript is as follow:

Figure 4. The design and overall performance of the cooling devices. c COP_{sys} , \dot{Q}_c , and \dot{W}_{in} as a function of T_{span} for the single-unit and double-unit cooling devices under the flow rate of 1.2 mL min^{-1} .

Question 4. *Regarding the specific heat capacity, the authors state, “The specific heat capacity (c_p) was confirmed by differential scanning calorimetry (DSC, TA-DSC2500) measurement with a heating rate of 10 K min⁻¹ in the temperature range of 203–443 K.” Is c_p obtained from a standard or modulated DSC measurement? How is it derived, considering that standard DSC alone is inaccurate for “ c_p ” characterisation?*

Response: Thank you for your in-depth comment. We strongly agree with you that the standard DSC alone is inaccurate for c_p characterization.

Actually, the c_p obtained in this work was obtained from a standard three-step DSC measurement. The c_p of the sample was calculated through the heat flow curve (P) of the baseline (blank run), sapphire (calibration run), and the sample (specimen run) through the following equation:

$$c_{p,\text{sample}} = c_{p,\text{calibration}} \frac{m_{\text{calibration}} (P_{\text{specimen run}} - P_{\text{blank run}})}{m_{\text{sample}} (P_{\text{calibration run}} - P_{\text{blank run}})}$$

The method for measuring the c_p of the sample using TA-DSC2500 is called the direct method developed by the TA Company. As you know, the DSC instrument needs to be calibrated before use. During the DSC instrument calibration steps, unlike other DSC instruments, commercial TA-DSC2500 has added Tzero calibration and direct heat capacity calibration steps in addition to the traditional T1 baseline calibration, furnace constant calibration, temperature calibration, and enthalpy calibration. This means that the mass and heat flow of the empty disk (blank run) and sapphire (specimen run) have been stored through these additional calibration steps. Consequently, the heat capacity and heat resistance of the sample end and the reference end inside the instrument can be obtained to minimize the instrument noise. After that, the c_p results of the sample are directly presented after being automatically analyzed through the built-in software during sample testing. Therefore, although the final data only presents a single curve (c_p of the sample), essentially it is still a three-step method.

To address this comment, we added the details of the so-called direct heat capacity calibration. The change in the revised supplementary information is as follows:

The specific heat capacity (c_p) was confirmed by the standard direct heat capacity measurement by using a differential scanning calorimetry (DSC, TA-DSC2500) instrument with a heating rate of 10 K min^{-1} in the temperature range of 203–443 K. Before testing the specific heat capacity, sapphire standard sample was used as the reference material for the Tzero and Direct heat capacity calibration of TA-DSC2500. Both calibration processes were performed with a heating rate of 10 K min^{-1} in the temperature range of 183–573 K.

(Line 13 of Page 6 in the Supplementary Information)

Question 5. *Timescales for heat transfer are rather long, which is a frequent problem in caloric cooling. This hinders operation frequency, which, in turn, constrains the device's cooling power. Figure S13, for instance, displays information on the water flow rate. More details should be given on the selection of the 0.167 Hz operation frequency and the performance on other frequencies.*

Response: Thank you for your valuable comment. To address this comment, we added the performance with different operating frequencies under a complete heat exchange condition and the calculated SCP values with different heat transfer timescales under an incomplete heat exchange condition in Supplementary Fig. 17 (Fig s13 in the first draft), and the corresponding notes on the frequency selection. The changes in the revised text are as follows:

(1) Text

When the thermal energy generated by a single operation of refrigerant is fully utilized, t_{cycle} is equal to the timescale for complete heat transfer (Supplementary Fig. 17a), so the frequency and t_{cycle} can be determined synchronously with the flow rate. In this case, all the caloric effect from the refrigerant can be utilized, and the system experienced the highest COP. Supplementary Fig. 17b shows the SCP of the cooling unit under a strain level from 100% to 500%. SCP reaches the maximum value of about 5.47 W g^{-1} when the V_{dot} is 1.2 mL min^{-1} . In addition, Supplementary Fig. 17c shows that the double-unit cooling device achieves the

highest cooling power and temperature span at a flow rate of 1.2 mL min^{-1} , which is mainly due to a balance between the amount and the duration of the heat exchange process. At this \dot{V}_{dot} , the heat transfer duration t_{cycle} is 6 s and the operating frequency is 0.167 Hz. Interestingly, increasing the cycling frequency when the thermal energy generated by a single operation of refrigerant is not fully utilized, the optimum t_{cycle} in terms of SCP can be less than the timescale for complete heat transfer. The theoretical Q_{out} and SCP values under different t_{cycle} and a constant flow rate of 1.2 mL min^{-1} can be calculated by integrating the water temperature change over different heat transfer timescales (Supplementary Fig. 17a). The theoretical COP under different t_{cycle} and a constant flow rate of 1.2 mL min^{-1} can be projected using the input work per cycle and the aforementioned integrated Q_{out} . As shown in Supplementary Fig. 17d, the calculated SCP reaches the maximum value of 9.7 W g^{-1} when the t_{cycle} is reduced to 2 s at a constant \dot{V}_{dot} of 1.2 mL min^{-1} . However, due to the incomplete heat exchange of the refrigerant, the motor consumption increases by about threefold, leading to a lower theoretical COP_{sys} . Therefore, considering COP_{sys} as a priority, in this study we choose 6 s as the t_{cycle} . A more comprehensive evaluation of the compromise between SCP and COP_{sys} should be conducted in the future.

(Line 4 of Page 16 in the Supplementary Information)

(2) Figure

Supplementary Fig. 17. Basic performance characteristics of the cooling devices and refrigerants. **a** ΔT of water versus time under different flow rates when the heat source is ambient. **b** Q_{out} and SCP under different flow rates when the heat source is ambient. **c** Cooling power versus temperature span under different flow rates and operation frequencies. **d** Calculated Q_{out} , SCP, and projected COP_{sys} under different heat transfer timescales when the flow rate is constant at 1.2 mL min⁻¹.

(Page 39 in the Supplementary Information)

Reviewer #3 (Remarks to the Author)

This study focuses on the elastocaloric effect of SEBS/graphene composites. It covers the development and characterisation of the materials as well as their integration into cooling systems. This work is interesting and could be of interest to the solid-state cooling community for the following reasons: 1) it highlights a significant elastocaloric effect; 2) until now, elastocaloric polymer composites have not been studied intensively; 3) the recovery capacity of the system has also been tested. However, the reviewer identified several shortcomings in the manuscript that should be corrected before reconsidering its publication:

Response: We thank you for your positive feedback. We also appreciate the literature you suggested and the comments which are helpful to improve the quality of this paper.

Question 1. *The title could mention that it concerns polymer composites.*

Response: Thank you for your kind suggestion. We have changed the title to address this comment:

Shearo-caloric **effect** enhanced elastocaloric **responses in polymer composites** for solid-state cooling

(Line 1 of Page 1 in the Main Manuscript)

Question 2. *The introduction to thermoplastic elastomers and composites is too short. Specifically, no work on the impact of deformation and the effect of carbon fillers on the structure and on the mechanical response of SEBS (or more generally on thermoplastic elastomers) is cited. However, there are a few works such as:*

Dechnarong, Macromolecules 2020, 53, 20, 8901-8909

Haonan Liu et al 2020 Jpn. J. Appl. Phys. 59 SN1013

Yongjin Li Macromolecules 2009, 42, 7, 2587-2593

Enrique-jimenez, European Polymer Journal Volume 97, December 2017, Pages 1-13

In addition, elastocaloric composites should also be clearly mentioned in the introduction.

The following references can be added and commented:

Nicolas Candau et al., Vol. 16, No.12, Pages 1331-1347, 2022

Seok Bin Hong et al 2019 Funct. Compos. Struct. 1 015004

Finally, the bibliography concerning polymeric elastocaloric systems is not clearly mentioned.

Response: Thank you for your comment. *To address this comment, we added the citations about the polymer composites, and elastocaloric system in the corresponding sections of the revised text. The changes are as follows:*

(1) Text

e-CEs have also been developed in polymer composites to take advantage of the reinforcement and nucleating ability, and high thermal conductivity of fillers.²⁶⁻²⁸

(Line 19 of Page 3 in the Main Manuscript)

(2) Text

On the other hand, solid-state cooling critically depends on functional development of the cooling devices that are tailored to solid refrigerant materials and their responses to the external fields.^{14-15, 32-37} However, the inherent large-deformation characteristics of polymers limit the development of their e-CE cooling devices due to the demand for large driving strokes. Only a few works have recently been reported on the large-deformation cooling polymer systems.^{21, 22, 29, 38} For example, by harnessing snap-through instability in the soft capacitor to drive the expansion and contraction of an NR membrane balloon, a high operating frequency function was achieved for the e-CE cooling device.²¹ Another type of large-deformation cooling device utilizes the geometric design of NR tubes to endow the device with a thermal conductivity compensation function.^{22, 38} Nevertheless, the energy recovery has not yet been realized in the

large-deformation systems, which is crucial for further improving the system efficiency.³⁹

(Line 4 of Page 4 in the Main Manuscript)

(3) Text

As illustrated in Fig. 2l, the **full-width** half maximum (FWHM) of the azimuthal (Ψ) integral curves for GNS/HHs after deformation gradually decreases with the GNSs loading increasing from 0 to 1 wt%.^{29,30,44}

(Line 6 of Page 8 in the Main Manuscript)

(4) Text

Film samples with a thickness of about 10 μm were prepared by solvent-casting of 25 g L⁻¹ nanofillers/SEBS slurries onto cleaned glass slides and used for the atomic force microscopy (AFM) test after drying at room temperature for seven days.^{2,3}

(Line 9 of Page 3 in the Supplementary Information)

(5) Reference

26. Candau, N. et al. Observation of heterogeneities in elastocaloric natural/wastes rubber composites. *Express Polym. Lett.* **16**, 1331-1347 (2022).
27. Colman, F. C. et al. On the mechanocaloric effect of natural graphite/thermoplastic polyurethane composites. *J. Mater. Sci.* **58**, 11029-11043 (2023).
28. Hong, S. B., An, Y. & Yu, W. R. Elastocaloric effects of carbon fabric-reinforced shape memory polymer composites. *Funct. Compos. and Struct.* **1**, 015004 (2019).
32. Defay, E., et al. Enhanced electrocaloric efficiency via energy recovery. *Nat. Commun.* **9**, 1827 (2018).
33. Li, J., et al. High cooling performance in a double-loop electrocaloric heat pump. *Science* **382**. 801-805 (2023).
34. Cui, H., et al. Flexible microfluidic electrocaloric cooling capillary tube with giant specific device cooling power density. *Joule* **6**, 258-268 (2022).
35. Ahčin, Ž., et al. High-performance cooling and heat pumping based on fatigue-resistant elastocaloric effect in compression. *Joule* **6**, 2338-2357 (2022).

36. Bruederlin, F., Ossmer, H., Wendler, F., Miyazaki, S. & Kohl, M. SMA foil-based elastocaloric cooling: from material behavior to device engineering. *J. Phys. D Appl. Phys.* **50**, 424003 (2017).
37. Kirsch, S. M. et al. NiTi-based elastocaloric cooling on the macroscale: From basic concepts to realization. *Energy Technol.* **6**, 1567–1587 (2018).
38. Sebald, G., et al. High-performance polymer-based regenerative elastocaloric cooler. *Appl. Therm. Eng.* **223**, 120016 (2023).
39. Qian, S. Thermodynamics of elastocaloric cooling and heat pump cycles. *Appl. Therm. Eng.* **219**, 119540 (2023).
44. Dechnarong, N. et al. In situ synchrotron radiation X-ray scattering investigation of a microphase-separated structure of thermoplastic elastomers under uniaxial and equi-biaxial deformation modes. *Macromolecules* **53**, 8901-8909 (2020).

(Page 17-18 in the Main Manuscript)

(6) Reference

2. Enrique-Jimenez, P. et al. Control of the structure and properties of SEBS nanocomposites via chemical modification of graphene with polymer brushes. *Eur. Polym. J.* **97**, 1-13 (2017).
3. Han, X., Hu, J., Liu, H. & Hu, Y. SEBS aggregate patterning at a surface studied by atomic force microscopy. *Langmuir* **22**, 3428-3433 (2006).

(Page 51 in the Supplementary Information)

Question 3. *Regarding material processing. It is written that the composites were agitated for 3 days. Did the authors ensure that there was no sedimentation of the particles?*

Response: Thank you for your question. There was no sedimentation of the particles under continuous stirring.

During the three-day stirring process, the solvent gradually evaporates, leading to an increase in solution viscosity and the formation of slurry. This ensures that even after removing

the stirring, the particles will not precipitate. We chose this processing method based on our past work on preparing asphalt and cement, which is also widely used in coal preparation and utilization. Similar methods have been used in the preparation of polymer composites (e.g., Jiang-Shan Gao et al. 2019 Results in Physics 15, 102720). We added this citation to the revised text:

(1) Text

After the matrix solutions were stirred to clear, the nanofiller solutions were proportionally added to the polymer solution and stirred for three days to prepare the nanofillers/SEBS slurries.¹

(Line 19 of Page 2 in the Supplementary Information)

(2) Reference

1. Gao, J. S., Liu, Z., Yan, Z. & He Y. A novel slurry bending method for a uniform dispersion of carbon nanotubes in natural rubber composites. *Results Phys.* **15**, 102720 (2019).

(Page 51 in the Supplementary Information)

Question 4. *Regarding the processing of the materials (in the additional information file). It is mentioned that the composites have a thickness that can vary between: 0.5 and 1 mm. ("And finally, sample sheets of nanofiller/SEBS composites (0.5-1 mm thick) were obtained"). This sentence raises two questions:*

For the solvent casting process, it is difficult to remove all the solvent when the thickness is greater than around 100-200 micrometers. Have the authors carried out a TGA (thermal gravimetric analysis) to check that no solvent is present inside the composites at the end of the process? If any residual solvent remains, what is the weight fraction of solvent for each formulation?

The second concern is that adiabatic conditions depend on the strain rate, thickness and thermal conductivity. The adiabatic deformation rate of 5 s⁻¹ mentioned in the supporting information concerns which thickness and which formulation? (thermal conductivity can be

affected by the presence of carbon fillers (man, F.C.. J Mater Sci 58, 11029-11043 (2023) and the convective heat transfer constant is related to thickness (Mott et al., PolymerVolume 105, 22 November 2016, Pages 227-233) which varies with the formulation in this work). These points are important because the filler optimum determined via DT_{adiab} measurements could be influenced by the way it is measured. (If the measurement is not adiabatic, the temperature variations cannot be compared).

Response: Thank you for your questions. *To address this comment*, we added more details to strengthen our results.

For the solvent residue, we added the TGA to confirm the solvent content. The results indicate that there are no solvent residues in the samples of this work.

For the adiabatic conditions. The thickness of the samples for e -CE measurement is all 0.5 mm. The change in the thermal conductivities, caused by the addition of fillers in our work, has no significant impact on the adiabatic conditions. The loading of fillers in Colman F. C.'s work is above the percolation threshold, which is significantly different from our work.

We also added the data on the surface $|\Delta T|$ of 1GNS/HHs as a function of strain rates to further demonstrate the adiabatic test condition. The results indicate that the strain rate of 15 s^{-1} used in this work still meets the adiabatic requirements.

The changes in the revised supplementary information are as follows:

(1) Method

Finally, the nanofillers/SEBS composites sample sheets (0.5-1 mm thickness) were obtained. After that, the thermalgravimetric analysis (TGA, NETZSCH STA 2500) was performed to give the thermogravimetric curves and derivative thermogravimetric (DTG) curves of the GNS/HHs (1 mm thickness). The results show that there is no weight loss related to solvent volatilization before the decomposition temperature of the matrix, which confirmed that there were no solvent residues in the sample sheets (details in Supplementary Fig. 4).

(Line 23 of Page 2 in the Supplementary Information)

(2) Method

TGA was performed with a heating rate of 10 K min^{-1} in the temperature range of 298–1273 K under the nitrogen atmosphere.

(Line 23 of Page 5 in the Supplementary Information)

(3) Method

Considering that the loading of carbon fillers may affect the thermal conductivity (see Supplementary Table 5) of the *e*-CE samples and lead to the differences in the adiabatic strain rates of the *e*-CE samples, direct measurement of the surface temperature changes ($|\Delta T|$) of HH, 1GNS/HH, and 8GNS/HH were performed in an open indoor environment at different strain rates. As shown in Supplementary Fig. 1, the tested $|\Delta T|$ s of all samples increase with the increase of strain rates when the strain rates are less than 5 s^{-1} . This is caused by the convection and radiation heat transfer between the test sample and the ambient environment at low strain rates. When the strain rates are greater than 5 s^{-1} , the $|\Delta T|$ values gradually become constant. This means that the heat transfer duration is shortened at higher strain rates, and a strain rate higher than 5 s^{-1} in an open indoor environment is equivalent to the adiabatic test condition. Since the adiabatic strain rate was estimated to be about 5 s^{-1} , the examination condition with a strain rate of 15 s^{-1} , which was far higher than the adiabatic strain rate, in an open indoor environment was selected to quantify ΔT_{adi} .⁴

(Line 18 of Page 4 in the Supplementary Information)

(4) Method

Thermal conductivity measurements were performed by the hot disk thermal analyzer (TPS2500S, Hot Disk Instrument) via the transient plate heat source method (ISO22007-2).

(Line 5 of Page 7 in the Supplementary Information)

(5) Figure

(Page 26 in the Supplementary Information)

(6) Figure

(Page 23 in the Supplementary Information)

(7) Table

Supplementary Table 5. Thermal conductivity of GNS/HHs at 298 K.

Sample	Thermal conductivity (W m ⁻¹ K ⁻¹)
HH	0.1673±0.0001
1GNS/HH	0.1960±0.0002
8GNS/HH	0.4853±0.0003

(Page 50 in the Supplementary Information)

Question 5. *Regarding the material processing for AFM, it is not the same as that used for the rest of the study. However, it is known that a skin effect can occur when using the solvent casting method (K. Wongtimnoi et al. , Volume136, Issue3, January 15, 2019) and my concern is that the AFM observation may not be representative of the bulk structure. I would recommend that the authors use the samples obtained with the conventional process (leading to a thickness of about 500µm) to prepare the surface of the sample for AFM by cryo ultra-microtomy.*

Response: Thank you for your comment. We agree with you that the method is not completely consistent with that used for the rest of the study and believe that the method used in this work to demonstrate the interaction of the PS-GNS microdomain structures is reasonable.

We have tried to prepare the AFM sample by using cryo ultra-microtomy. However, the maximum range of thickness produced by the commercial slicer is 10 nm to 15 µm. The commercial slicer cannot meet the recommendation of 500 µm. Moreover, the size of the sliced sample is micron-sized, whereas the maximum scanning range of AFM used in this work is 1 micron. Since the imaging process of AFM involves continuous point-by-point tapping on a centimeter-sized sample holder, rather than the direct observation, searching for interaction areas in the samples is challenging. A more challenging issue is that the height difference between the sample and the sample holder obtained by cryo ultra-microtomy exceeds the

maximum fluctuation range of AFM requirement, which makes the measurement impossible to complete. For these reasons, we did not choose this sampling method.

The processing method was chosen after careful consideration on the strict sampling requirements for AFM phase diagram measurements. For AFM testing, the skin effect will only affect the testing of surface roughness and not affect the testing of phase diagrams within the acceptable sample fluctuation range of the instrument. Phase diagrams as an important extension technique of tapping modes, are imaged by detecting the difference between the phase angle of the signal source driven microcantilever probe vibration and the actual phase angle of the microcantilever probe vibration (*i.e.* the phase shift between the two). The factors causing this phase shift come from the composition, hardness, and modulus. This is exactly what we need to determine the polystyrene domain, GNS, and PEB domain.

The selected sample thickness for the AFM test in this work mainly comes from the compatibility with the sample preparation requirements for phase diagram measurement. A thin and flat sample is required for the phase diagram measurements (Langmuir 2006, 22, 3428-3433). Moreover, the AFM phase diagrams in this work were only used to support the influence of GNSs on the microdomain structures of the composites. This influence comes from the interactions between components, which would not be related to whether the sample is a bulk or surface. The solvent-casting components of AFM samples in this work are the same as that used for the rest of the study, therefore it is reasonable to illustrate the microdomain structure. Similar methods have also been used in the other works, such as mentioned in your Question 2 (the Preparation of nanocomposites section and Fig 8 in P. Enrique-Jimenez's work.).

We added these two citations in the revised text. The changes are as follows:

(1) Text

Film samples with a thickness of about 10 μm were prepared by solvent-casting of 25 g L^{-1} nanofillers/SEBS slurries onto cleaned glass slides and used for the atomic force microscopy (AFM) test after drying at room temperature for seven days.^{2,3}

(Line 9 of Page 3 in the Supplementary Information)

(2) Reference

2. Enrique-Jimenez, P. et al. Control of the structure and properties of SEBS nanocomposites via chemical modification of graphene with polymer brushes. *Eur. Polym. J.* **97**, 1-13 (2017).
3. Han, X., Hu, J., Liu, H. & Hu, Y. SEBS aggregate patterning at a surface studied by atomic force microscopy. *Langmuir* **22**, 3428-3433 (2006).

(Page 51 in the Supplementary Information)

Question 6. *In the results and discussion section, P5 lines 4-5, the authors should clearly mention: the molecular chain length, side group content and weight fraction of the PS/PEB so that the reader can easily relate it to their previous study. They should also provide references to the polymers supplied by KRATON and Northwest Rubber & Plastics Research & Design Institute Co, Ltd in the supplementary informations. They should comment on Figure S1 and indicate the molecular weight in supplementary section. For HH and LH, the molecular weight is not readable in Figure S1.*

Response: Thank you for your comment. *To address this comment, we added the average molecular weight, side group content, and weight fraction of polystyrene in the revised manuscript. We also added the website link to the revised supplementary information. The molecular weight is added in the legend of Supplementary Fig. 2 (Figure S1 in the first draft).*

The changes in the revised manuscript and supplementary information are as follows:

(1)

The fabrication and labeling of the polymer composites are summarized in **Supplementary Methods, Supplementary Table 1, and Supplementary Figs. 2-4**. The SEBSs with high molecular chain-length uniformity²⁹ and high orientation ability³⁰ are labeled as HHs. **The molecular weight distribution range, the side group content, and the weight fraction of polystyrene (PS) of HH are 79,000 g mol⁻¹, 11 mol%, and 29 wt%, respectively.**

(Line 11 of Page 5 in the Main Manuscript)

(2)

Three kinds of triblock poly(styrene-*b*-ethylene-*co*-butylene-*b*-styrene) (SEBS) thermoplastic elastomers, with polystyrene (PS) hard blocks surrounding a poly(ethylene-*co*-butylene) (PEB) central soft block, used in this study were provided by KRATON Polymers Co., Ltd. (G1650MV, <http://www.kraton-polymers.cn/>), TSRC Industries Co., Ltd. (6159, <https://www.tsrc.com.tw/>), and Northwest Rubber & Plastics Research & Design Institute Co., Ltd., China (as-40, <http://www.xbxj.chemchina.com/xbxjy/index.htm>).

(Line 3 of Page 2 in the Supplementary Information)

(3)

Supplementary Figure 2. GPC curves of composite matrices. The maximum M_{w-max} appears at $H_t=90\%$. The minimum M_{w-min} appears at $H_t=10\%$. The weight average molecular weight of HH, LH, HL are $100,921 \text{ g mol}^{-1}$, $1,271,743 \text{ g mol}^{-1}$, and $129,698 \text{ g mol}^{-1}$, respectively

(Page 24 in the Supplementary Information)

Question 7. *In the results and discussion section, p5 line 8, I don't understand what "ultimate strains" means (although this point seems important for comparing DT_{adi})? Is it just below the strain at break? Is it 600% for each sample (Figure 1.b)? Is it also possible to add adiabatic temperature values for different strains?*

Response: Thank you for your question. The ultimate strain in this work was defined as the former strain before the fracture strain of a sample. A strain of 600% is for each sample shown in Fig. 1a. *To address this comment*, we made annotations in the main text and added the adiabatic temperature change of each GNS/HHs under different strains. The changes are as follows:

(1)

As shown in Fig. 1a, the highest $|\Delta T_{adi}|$ value of GNS/HHs **in the** cooling process, which was

measured under the largest strain (ϵ) of 600%, ... The $|\Delta T_{\text{adi}}|$ values of the selected GNS/HHs at different strain levels during the cooling process are summarized in Supplementary Fig. 5. (Line 16 of Page 5 in the Main Manuscript)

(2)

Figure 1. e-CE cycle and temperature change of GNS/HHs. a $|\Delta T_{\text{adi}}|$ of GNS/HHs on cooling process. All data were collected at a strain level of 600%.

(3)

(Page 27 in the Supplementary Information)

Question 8. In the results and discussion section, for the sentence "This room temperature e-CE activity exceeded those of previously reported elastocaloric polymers (Fig. s3)" the authors should also refer to the work of Greibich et al. which claims a temperature variation of around 23K (Greibich, F et al. Nat Energy 6, 260-267 (2021).) and also the work of Run Wang et al (SCIENCE Oct 2019 Vol 366, Issue 6462 pp. 216-221) and perhaps adjust this sentence . They could also adapt the sentence in the abstract: "The consequential adiabatic temperature change of -18.0 K and isothermal entropy change of $187.4 \text{ J kg}^{-1} \text{ K}^{-1}$ exceed those of other

elastocaloric polymers."

Response: Thank you for your comment. We believe that our conclusions in the discussion section and abstract are reasonable because of the following reasons.

Firstly, the thermal change of the elastocaloric effect is induced by the change of magnitude of a uniaxial stress (Mañosa, L. & Planes, A. Materials with giant mechanocaloric effects: cooling by strength. *Adv. Mater.* **29**, 1603607 (2017)). However, these two works mainly discussed the caloric effects generated during isotropic deformations (Grebich's work) and twist changes (Run's work).

Secondly, the most eye-catching part of Grebich's work for us is their clever device design and unusual driving methods, demonstrating extremely high device performance. At the material level, the temperature variation of around **23 K** of Grebich's work refers to the temperature variation between step 2 and 4, which is not the adiabatic temperature change of caloric effect response ΔT_{adi} . The ΔT_{adi} of Grebich's work would be about **8 K**, which refers to the temperature variation between the temperature at step 3 and 4 (fig 1c in Grebich's work). In other word, the temperature variation in our work would be **43.1 K** using Grebich's calculation method. The temperature variation in our work is higher than that of Grebich's work in both comparison methods.

Thirdly, the ΔT_{adi} of the elastocaloric effect of NR in Run's work is **12.2 K** (fig 1b in Run's work). The maximum ΔT_{adi} superimposed by both twistocaloric and elastocaloric effects is **14.4 K** (fig. 2E in Run's work). The ΔT_{adi} in our work is also higher than that in Run's work.

The elastocaloric responses of NR described in the works of Grebich and Run are both added to the caloric responses range of NRs summarized in Supplementary Fig. 6 (fig. s3 in the first draft). The changes in the revised text are as follows:

(1) Figure

(Page 28 in the Supplementary Information)

(2) Reference

19. Wang, R. et al. Torsional refrigeration by twisted, coiled, and supercoiled fibers. *Science* **366**, 216-221 (2019).
20. Greibich, F. et al. Elastocaloric heat pump with specific cooling power of 20.9 W g⁻¹ exploiting snap-through instability and strain-induced crystallization. *Nat. Energ.* **6**, 260-267 (2021).

(Page 52 in the Supplementary Information)

Question 9. Figure. 2, the strain at which the characterisation was carried out is not mentioned. This point is really important especially for figure.2.k. It might also be interesting to see how it evolves for each formulation with strain since in situ SEBS study shows that it is not monotonous (Dechnarong, *Macromolecules* 2020, 53, 20, 8901-8909). Besides, how long after deformation was the Xray measurement taken? (Was the stress stable?) For figures 2.h) and 2.i),

why the maximum of intensity is lower after stretching for 1GNS/HH (green in the colour scale) than for HH (yellow in the colour scale)? My concern is that the peak should be thinner and higher for 1GNS/HH (not only thinner).

Response: Thank you for your suggestions on WAXD.

To address the comments on the test details, we added annotations related to strain levels in the figures and captions in Fig. 2. The 2D WAXS measurements were taken immediately after the sample reached the predetermined strain. The stress was not stable due to the stress relaxation phenomenon in polymers during the duration of the exposure process of WAXS. The exposure duration was set as 120 s for each picture, which is the same as that for heat exchange. We added these details in the revised text. The in-situ 2D WAXS study of Dechnarong's work shows that the evaluation is monotonous (refers to Figs. 9 and 10 in Dechnarong's work), and this monotonous evaluation has also been proven in other studies (Fig. s6 in doi.org/10.1038/s41467-021-27746-y). The 2D WAXS used in this work is to compare and demonstrate the maximum structural (orientation) differences between GNS/HHs, which is independent of other strain levels.

*For the test results, the maximum of the cumulative intensity is lower for 1GNS/HH is reasonable, due to the thickness difference of HH and 1GNS/HH. More molecular chains in the 3 (beam direction) and 2 directions of 1GNS/HH sample could participate in orientation along the 1 direction (stretching direction) due to the shear effect of GNS. This resulted in the 1GNS/HH sample becoming thinner in the 3 direction. The signal cumulative intensity of WAXS is extremely sensitive to the irradiated volume, thus, the thinner sample exhibits lower signal intensity under the same testing condition. For this reason, the peak should be thinner and lower for 1GNS/HH. This is also the reason that the absolute signal cumulative intensity cannot be directly compared between two samples in WAXS. The conventional approach is to compare the relative intensity via azimuthal integration, *i.e.*, Fig. 21.*

We added more notes on the method. The changes in the revised text are as follows:

(1)

Figure 2. ... **k** 1D WAXD profiles of GNS/HHs obtained by full-titled circular integration of the corresponding 2D WAXD patterns at P1 ($\epsilon = 0\%$). **l** Azimuthal integral of the corresponding 2D WAXD patterns of GNS/HHs at P2 ($\epsilon = 600\%$) obtained near $q = 1.3 \text{ \AA}^{-1}$. The inset profile is a **partially** enlarged view of Fig. 2l. The black dotted line indicates the position of half maximum.

(2)

The 2D WAXS measurements were taken immediately after the sample reached the predetermined strain. The exposure duration was set as 120 s for each picture.

(Line 6 of Page 6 in the Supplementary Information)

(3)

(4)

Figure 2. The structure evolution and the enhancement mechanism of s-CE for GNS/HHs. h-i 2D WAXD images of HH (h) and 1GNS/HH (i) under P1 and P2.

(5)

Figure 2. The structure evolution and the enhancement mechanism of s-CE for GNS/HHs. j Magnification image for 1GNS/HH under P2 with enhanced contrast. The white arrows point to the (002) crystal plane of GNS.

Question 10. *Figure.2 , lower magnification images could be added in TEM and AFM. From AFM observation (Figure 2.d), it appears that the composites are made up of huge aggregates and therefore the number of chains interacting with the fillers should not be high and perhaps*

not in agreement with Figure 2.a) which suggests that most chains interact with the fillers.

Response: Thank you for your comment. To address this comment, lower magnification images of TEM and AFM have been added in the revised text. We also made modifications to the minimum elastic units in the schematic diagram to better describe the shear layer and the interlaminar molecular chains. The changes are as follows:

(1)

Lower magnifications of the TEM image and AFM phase diagram of 1GNS/HH are displayed in Supplementary Fig. 10.

(Line 11 of Page 7 in the Main Manuscript)

(2)

(Page 32 in the Supplementary Information)

(3)

Question 11. With regard to end-to-end determination, is it possible to use the Herman function directly in nanocomposites of thermoplastic block polymers as you did or is it only possible in homopolymers without fillers? Please mention other work using this function in block copolymer composites.

Response: Thank you for your question. The Herman's equation is a universal qualitative method for analyzing the orientation degree of the structure corresponding to the data.

Essentially, this equation ($f = \frac{1}{2}(3 \cos^2(\Psi - 90^\circ) - 1)$) is only related to the structure described in the data, which is independent of whether the sample is a block polymer or composite.

The structure, that dominates the *e*-CE response in this work, is the PEB segment. The scattering vector corresponding to this structure (PEB segment) is $q = 1.3 \text{ \AA}^{-1}$. While those corresponding to PS and GNS are about 0.0232 \AA^{-1} (supplementary discussion 2 in doi.org/10.1038/s41467-021-27746-y) and 1.8 \AA^{-1} , respectively. As illustrated in the caption

of Fig. 2l, the data here was obtained near $q = 1.3 \text{ \AA}^{-1}$. Therefore, it is reasonable to use this function in this work.

We added citations where relevant in the text:

(1) Text

As illustrated in Fig. 2l, the **full-width** half maximum (FWHM) of the azimuthal (Ψ) integral curves for GNS/HHs after deformation gradually decreases with increasing the GNSs loading from 0 to 1 wt%.^{29,30,44}

(Line 6 of Page 8 in the Main Manuscript)

(2) Reference

44. Dechnarong, N. et al. In situ synchrotron radiation X-ray scattering investigation of a microphase-separated structure of thermoplastic elastomers under uniaxial and equi-biaxial deformation modes. *Macromolecules* **53**, 8901-8909 (2020).

(Page 18 in the Main Manuscript)

Question 12. *With regard to end-to-end determination and affine deformation:*

how is Figure.2.m obtained? The authors should mention in the main text that it is extracted from the statistical study presented in the supplementary information (which is carried out under the assumption of affine deformation).

Can the affine deformation assumption and entropy elasticity formula (equ s2 , S5 s6) be used for thermoplastic elastomers composites? Authors may cite papers to support this, because it looks like to be in disagreement with “Dechnarong ,Macromolecules 2020, 53, 20, 8901-8909” since it was shown that a non affine deformation occurs for SEBS polymers above a critical strain and also because plasticity is observed in figures.s7 and figure.s13.

Response: Thank you for your questions.

Fig. 2m was obtained from the GNS/HHs models as presented in the beginning of the paragraph. To address this comment, we added additional notes in the revised texts.

And thank you for your suggestion on the Statistical Theory of Rubber Elasticity. The affine deformation is one of the fundamental assumptions for the Statistical Theory of Rubber Elasticity, which can apply to any polymer that displays rubber-like behavior. We added citations in the revised text.

The deviation from affine deformation in Dechnarong's work, which started in the large deformation region, is mainly related to another assumption (Gaussian chain assumption) of the Statistical Theory of Rubber Elasticity. The molecular chains in the large deformation region will cause the actual end-to-end distance to deviate from that of the Gaussian chains, resulting in a stress enhancement. This phenomenon occurs in all polymer networks and is regardless whether or not the system is a thermoplastic elastomer composite.

The deviation from affine deformation caused by the non-affine deformation mainly originates from the fluctuation of crosslinking points, which also occurs in all polymer networks. In fact, the enhancement of the e -CE in this work comes from the reduction of the number of non-radiative molecular chains by fillers. The plastic deformation in this work, similar to the Mullins effect, can be eliminated after multiple elongation-recovery cycles (10 times). Even though there are deviations in the description of actual networks using the Statistical Theory of Rubber Elasticity, it does not affect its correctness as an ideal eigenmodal to demonstrate the conformational change of molecular chains during deformation. Therefore, it is reasonable to use it to explain the energy conversion associated with conformational changes during polymer deformation.

The changes in the revised texts are as follows:

(1) Text

The above process was further validated through the dynamic simulation (details in **Supplementary Methods**). ... The statistic images (**Fig. 2m**) for the end-to-end distance of **the above GNS/HHs models** show that the molecular chains of GNS/HHs possess a similar mean square end-to-end distance value of about 4054 \AA^2 to HH at the initial state.

(Line 18 of Page 8, Line 2 of Page 9 in the Main Manuscript)

(2) Text

When the molecular chains are Gaussian chains, and the motion of crosslinking points conforms to affine deformation, the conformational entropy change ΔS_λ can be further quantified by the statistical theory of rubber elasticity^{12, 13}:

(Line 18 Page 8 in the Supplementary Information)

(3) Reference

12. Wall, F. T. Statistic thermodynamics of rubber. *J. Chem. Phys.* **10**, 132-134 (1941).

13. Flory, P. J. & John, R. J. Statistical mechanics of cross - linked polymer networks I. Rubberlike elasticity. *J. Chem. Phys.* **11**, 512 (1943).

(Page 51 in the Supplementary Information)

Question 13. *In the paragraph "Structure evolution and enhancement mechanism of CE for GNS/HHs", no references are cited to support and/or compare the conclusions, i.e. for example, references relating to the impact of stress on the stretching of macromolecules and on the interaction of GNS with PS blocks could be added.*

Response: Thank you for your comment. *To address this comment, we added corresponding references. The change is as follow:*

(1) Text

In addition, the π - π interactions formed between the polystyrene domains in the polymer and the GNS nanofillers enhance the interfacial adhesion between the nanofillers and the polymer matrix.^{42, 43}

(Line 21 of Page 6 in the Main Manuscript)

(2) Reference

42. Shen, B. et al. Melt blending in situ enhances the interaction between polystyrene and graphene through π - π stacking. *ACS Appl. Mater. Interfaces* **3**, 3103–3109 (2011).

43. Hu, K., Kulkarni, D. D., Choi, I. & Tsukruk, V. V. Graphene-polymer nanocomposites for structural and functional applications. *Prog. Polym. Sci.* **39**, 1934-1972 (2014).

(Page 18 in the Main Manuscript)

Question 14. *Page 9 lines 14-18: "With the increase in GNS loading from 0 to 1 wt%, the maximum values of $\tan \delta$ corresponding to T_g -PEB progressively decreased, and those corresponding to T_g -PS progressively increased". In this case, why for 5 and 8% of GNS loading, the peak associated with PEB T_g is higher and that associated with PS T_g is lower than the unloaded HH matrix and why the rubbery plateau is much lower for 8 GNS/HH. Furthermore, the authors should probably refer to the alpha relaxation temperature rather than the glass transition temperature. Finally, they could perhaps use literature such as the work of S. Kuester et al / Composites Part B 84 (2016) 236-247 to compare and interpret their data.*

Response: Thank you for your comments. For the DMA data. Essentially, the storage modulus and $\tan \delta$ represent the elasticity and the vibration absorption of the specific segments, respectively. *To address this comment*, we proposed a more specific concept of shear-damping transition caused by the filler loading. We also added more detailed explanations and citations related to the principles of DMA to increase readability.

For the definition, the α relaxation corresponds to the main transitions in polymers, which is related to the segment motion of a specific domain. The temperature at the α relaxation corresponds to different transition temperatures in different polymer systems. For example, T_c and T_m represent crystallization and melting transitions, respectively. In this work, the α relaxation refers to the glass transition of PEB and PS domains. T_g -PEB and T_g -PS correspond to the glass-rubber transition and the rubber-viscous transition temperatures of the polymer composite, respectively.

Thanks for the suggested literature. The DMA data shown in S. Kuester's work is consistent with those for GNS loading from 0-1 wt% in this work.

The changes in the revised text are as follows:

(1) Text

Shear-damping transition in different composite systems

As shown in Fig. 1a, $|\Delta T_{\text{adi}}|$ exhibits a progressive rise with the GNS loading and reaches its peak value around 1 wt.% loading (*i.e.*, the critical loading) of GNS. With further increasing the GNS content beyond the critical loading, the decrease of $|\Delta T_{\text{adi}}|$ primarily arises from the occurrence of a shear-damping transition within the reconstructed network.

As illustrated in Fig. 3a, the $\tan\delta$ peak near 235 K is attributed to the glass transition temperature (T_g) of the PEB elastic blocks ($T_{g\text{-PEB}}$), and the peak at near 371 K is assigned to the T_g of the polystyrene hard blocks ($T_{g\text{-PS}}$). Before reaching the critical loading, the peak value of $\tan\delta$ corresponding to the $T_{g\text{-PEB}}$ gradually decreases with the GNS content, while the peak assigned to the $T_{g\text{-PS}}$ steadily increases. This indicates that the fraction of the polymer segments affected by the interfacial bonding of GNS is significantly increased, which endows the composite with more *s*-CE elastic elements, specifically, higher storage modulus under rubber state (Fig. 3b). It can be considered that the viscoelastic behavior of the reconstructed network, which is caused by the introduced GNS, is mainly reflected by the increased fraction of the *s*-CE elastic elements before the critical loading. The increase of the *s*-CE elastic elements reduces the content of viscous fraction and transmits stress throughout the entire cross-linking composite, thereby attenuating the damping performance of PEB soft domains and manifesting a reduced $\tan\delta$ associated with the $T_{g\text{-PEB}}$. On the other hand, the physical entanglements of the polystyrene phase is weakened due to the additional π - π interactions between polystyrene segments and GNS, resulting in higher segmental mobility of polystyrene during its glass transition process and leading to an enhanced $\tan\delta$ associated with the $T_{g\text{-PS}}$.⁴⁵ Therefore, the degree of orientation (Fig. 2l) and the resulting *e*-CE (Fig. 1a) increase with the increase of GNS loading up to 1 wt%.

With the further increase of GNS loading (> 1 wt%), both the degree of orientation of the deformed composite and the obtained *e*-CE decrease. This is owing to the fact that the partially reconstructed network could undergo a shear-damping transition and exhibit higher damping performance (Fig. 3a) when the GNS content is beyond the critical loading. The increased loading of GNS reduces the interparticle distance and forms nanofiller aggregates as verified in the field emission-scanning electron microscopic (FE-SEM) images (Supplementary Fig.

14). The formation of the nanofiller **aggregate** network greatly reduces the relative movement between GNSs and the associated shear effects, **which means that the deformation or destruction of GNS aggregates under dynamic force would lead to more friction and viscosity of the embedded PEB segments.**⁴⁶ The nanofiller aggregates can be considered as damping elements and contribute to the vibration absorption behavior of the reconstructed network. Thus, the deformation or destruction of the nanofiller aggregate network would significantly increase the $\tan\delta$ near the T_g -PEB and **result** in a lower storage modulus in **the** rubber state (Fig. 3b). **Moreover,** a large amount of GNSs may refine the domain size of **the** polystyrene phase. **Consequently, a higher fraction of the polystyrene segments could interact with GNS, thereby reducing the mobility of polystyrene segments and** resulting in smaller $\tan \delta$ near the T_g -PS.

(Page 9-11 in the Main Manuscript)

(2) Reference

45. Kueser, S. et al. Processing and characterization of conductive composites based on poly(styrene-*b*-ethylene-*ran*-butylene-*b*-styrene) (SEBS) and carbon additives: A comparative study of expanded graphite and carbon black. *Compos. Part B-Eng.* **84**, 236-247 (2016).

46. Zeng, X. et al. Elastomer composites with high damping and thermal resistance via hierarchical interactions and regulating filler. *Small* 2306946 (2023).

(Page 19 in the Main Manuscript)

Question 15. *Regarding Table S1 and Figure S4, the reviewer does not understand why so much data is presented for carbon black and carbon nanotube composites, since structure and temperature variation are only evaluated for "GNS/HH" composites in the main part of the paper.*

Response: Thank you for your question. The initial intention was to demonstrate the universality of the *s*-CE reinforcement in different matrix-filler systems.

To address this comment, we added additions notes in the revised texts:

(1)

Interestingly, the *s*-CE and the shear-damping transition have been found in other nanofiller-polymer composites (details in Supplementary Discussion 3 and Supplementary Fig. 15), demonstrating the generality of *s*-CE. It is demonstrated that the *s*-CE could enhance the original *e*-CE of the polymer matrix, with weak chain orientation mobility, by 30% at most. Consistent to the results of the 1GNS/HH composite, the shear-damping transition of the composites filled with a high nanofiller aspect ratio occurs at a low nanofiller loading.

(Line 7 of Page 11 in the Revised Main Manuscript)

(2)

Among them, GNS/HHs and MCNT/HHs showed a high *s*-CE at a lower loading, which may be due to the higher aspect ratio of MCNT and GNS compared to CB. Nanofillers with higher aspect ratios could connect more *s*-CE units, but it could also lead to a more easily formed nanofiller aggregate network. Hence, the onset of the shear-damping transition would occur at a reduced nanofiller loading.

(Line 15 of Page 13 in the Supplementary Information)

Question 16. *In the Double-Unit cooling device with work capacity recovery section, the authors could compare the performances of their system with those of Greibich et al (Greibich, F et al. Nat Energy 6, 260-267 (2021)), Sebald et al (Sebald et al. Applied Thermal Engineering Volume 223, 25 March 2023, 120016) and Run Wang et al (SCIENCE Oct 2019 Vol 366, Issue 6462 pp. 216-221).*

Response: Thank you for your comment. To address this comment, we added the comparison of the performances of these works. The changes in the revised text are as follows:

(1) Text

As illustrated in Supplementary Table 4, when the heat transfer medium is water and the shape

of the refrigerant is bulk tube,^{26,30} the low thermal conductivity further hinders the operation frequency and specific cooling power of the associated polymer-based cooling devices in order to ensure adequate heat exchange. In this case, the polymer-based cooling device could still maintain a higher temperature span. In this work, water flow is used as a heat transfer medium to extract the cooling energy generated from bulk refrigerant. As another major advantage of this configuration being the scalability of the refrigerant, it is possible to increase the mass of refrigerant, thereby increasing the cooling power, without significantly altering the device structure. Another configuration of the cooling device uses a thinner refrigerant to transfer heat through solid-solid contact.^{20,28} This configuration can significantly reduce the time required for heat transfer because of high specific surface area, thereby achieving higher operating frequency and specific cooling power. This makes the operation frequency of polymer-based cooling devices comparable to that of alloy-based devices while achieving a specific cooling power of up to 20.9 W g⁻¹. However, the cooling power of the polymer-based cooling device is much lower under this configuration. This is because specific cooling power is still related to both operation frequency and sample shape (specific surface area). As the value of specific cooling power does not take into consideration of the mass or volume factors associated with the change of specific surface area, any changes in heat transfer caused by the mass/volume factors or scalability have not been considered. In summary, to improve the overall performance of polymer-based cooling devices, it is necessary to further combine the reduction of the heat transfer period with the increase of refrigerant mass.

(Line 9 of Page 21 in the Supplementary Information)

(2) Table

Supplementary Table 4. Performance comparison of *e*-CE cooling systems and materials.

Sample	Status	Refrigerant			System						Ref.
		$\Delta\sigma$ (MPa)	ε (100%)	Cycles	Medium	COP_{sys}	T_{span} (K)	f (Hz)	\dot{Q}_c (W)	SCP (W g ⁻¹)	
NiTi	Membrane Sheet	450	0.034	2×10 ³	Water	3.5	15.3	0.25	4.64	0.8	27
TiNiFe	Membrane Foil	500	0.055	–	Cu metal	3.2	13	4	7.9	7.7	28
NiTi	Membrane Sheet	–	0.043	6×10 ³	Water	–	19.9	–	–	–	29
NiTi	Bulk tube	1234	–	10 ⁷	Water @	–	5.6	1.2	7.9	6.27	30
NiTi	Bulk tube	700	0.035	7.5×10 ⁴	Water	6.85	22.5	0.071	260	0.3	31
NR	Bulk fiber	–	0-1 #	750	Water	–	0.7 ##	–	–	–	19
NR	Membrane balloon	*	**	10 ³	Al metal	–	7.9 ***	1.1	0.75	20.9	20
NR	Bulk tube	1.5	3.5-5.5	3×10 ⁴	Water	6	8.3	0.1	1.5	0.14	26
1GNS/ HH	Bulk film	4	1-5	10 ³	Water	8.3	3.7	0.167	5.0	5.47	This work

T_{span} is collected near zero \dot{Q}_c . \dot{Q}_c and COP_{sys} are collected near zero T_{span} .

@ The heat is transferred by evaporation and condensation of water.

Accompanying by isometrical twisting and untwisting at 15 turns/s. ## Assume that the temperature drop of the water is the T_{span} under zero \dot{Q}_c .

* 8.5 kPa of gas pressure. ** Balloon volume change from 115 to 200 cm³. *** Temperature span between the hot and cold sides of a device.

(Page 49 in the Supplementary Information)

(3) Reference

19. Wang, R. et al. Torsional refrigeration by twisted, coiled, and supercoiled fibers. *Science* **366**, 216-221 (2019).
20. Greibich, F. et al. Elastocaloric heat pump with specific cooling power of 20.9 W g⁻¹

exploiting snap-through instability and strain-induced crystallization. *Nat. Energ.* **6**, 260-267 (2021).

26. Sebald, G., et al. High-performance polymer-based regenerative elastocaloric cooler. *Appl. Therm. Eng.* **223**, 120016 (2023).

(Page 52 in the Supplementary Information)

Question 17. *In the Double-Unit cooling device with work capacity recovery section, The massic cooling power (cooling power divided by mass of active materials) could also be presented to facilitate comparison between literature studies.*

Response: Thank you for your comment. The specific cooling power (SCP) at zero- T_{span} was 5.47 W g^{-1} when the flow rate of heat transfer water was 1.2 mL min^{-1} and the sample thickness was 1 mm within a strain range from 100% to 500% (Supplementary Fig. 17b). *To address this comment*, we added the comparison of SCP and the related references. The changes are as follows:

(1) Text

In this work, water flow is used as a heat transfer medium to extract the cooling energy generated from bulk refrigerant. As another major advantage of this configuration being the scalability of the refrigerant, it is possible to increase the mass of refrigerant, thereby increasing the cooling power, without significantly altering the device structure. Another configuration of the cooling device uses a thinner refrigerant to transfer heat through solid-solid contact.^{20,28} This configuration can significantly reduce the time required for heat transfer because of high specific surface area, thereby achieving higher operating frequency and specific cooling power. This makes the operation frequency of polymer-based cooling devices comparable to that of alloy-based devices while achieving a specific cooling power of up to 20.9 W g^{-1} . However, the cooling power of the polymer-based cooling device is much lower under this configuration. This is because specific cooling power is still related to both operation frequency and sample shape (specific surface area). As the value of specific cooling power does not take into

consideration of the mass or volume factors associated with the change of specific surface area, any changes in heat transfer caused by the mass/volume factors or scalability have not been considered. In summary, to improve the overall performance of polymer-based cooling devices, it is necessary to further combine the reduction of the heat transfer period with the increase of refrigerant mass.

(Line 13 of Page 21 in the Supplementary Information)

(2) Table

Supplementary Table 4. Performance comparison of *e*-CE cooling systems and materials.

Sample	Status	Refrigerant			System						Ref.
		$\Delta\sigma$ (MPa)	ε (100%)	Cycles	Medium	COP_{sys}	T_{span} (K)	f (Hz)	\dot{Q}_c (W)	SCP (W g ⁻¹)	
NiTi	Membrane Sheet	450	0.034	2×10^3	Water	3.5	15.3	0.25	4.64	0.8	27
TiNiFe	Membrane Foil	500	0.055	–	Cu metal	3.2	13	4	7.9	7.7	28
NiTi	Membrane Sheet	–	0.043	6×10^3	Water	–	19.9	–	–	–	29
NiTi	Bulk tube	1234	–	10^7	Water @	–	5.6	1.2	7.9	6.27	30
NiTi	Bulk tube	700	0.035	7.5×10^4	Water	6.85	22.5	0.071	260	0.3	31
NR	Bulk fiber	–	0-1 #	750	Water	–	0.7 ##	–	–	–	19
NR	Membrane balloon	*	**	10^3	Al metal	–	7.9 ***	1.1	0.75	20.9	20
NR	Bulk tube	1.5	3.5-5.5	3×10^4	Water	6	8.3	0.1	1.5	0.14	26
1GNS/ HH	Bulk film	4	1-5	10^3	Water	8.3	3.7	0.167	5.0	5.47	This work

T_{span} is collected near zero \dot{Q}_c . \dot{Q}_c and COP_{sys} are collected near zero T_{span} .

@ The heat is transferred by evaporation and condensation of water.

Accompanying by isometrical twisting and untwisting at 15 turns/s. ## Assume that the temperature drop of the water is the T_{span} under zero \dot{Q}_c .

* 8.5 kPa of gas pressure. ** Balloon volume change from 115 to 200 cm³. *** Temperature span

between the hot and cold sides of a device.

(Page 49 in the Supplementary Information)

(3) Reference

19. Wang, R. et al. Torsional refrigeration by twisted, coiled, and supercoiled fibers. *Science* **366**, 216-221 (2019).
20. Greibich, F. et al. Elastocaloric heat pump with specific cooling power of 20.9 W g⁻¹ exploiting snap-through instability and strain-induced crystallization. *Nat. Energ.* **6**, 260-267 (2021).
26. Sebald, G., et al. High-performance polymer-based regenerative elastocaloric cooler. *Appl. Therm. Eng.* **223**, 120016 (2023).

(Page 52 in the Supplementary Information)

Question 18. *The recovery capacity is really interesting. This makes sense of the COP_{mat} in which the heat that can be exchanged is divided by the mechanical hysteresis assuming that the unloading work can be reused (Cui et al. Applied Physics Letters 101, 073904 (2012)). The authors could try to better explain the link between the mechanical hysteresis of the material (outside the system) and the input work which is required in the system. The ideal consumed energy and the real consumed energy are not clear for me, so please improve the quality of the explanation of equ s18. Besides, the authors could check the literature for other systems that prove the possibility of using recovery work and mention the associated literature if it exists, otherwise I recommend them to better highlight this part of their work if it is new.*

Response: Thank you for your suggestions on the recovery capacity and the literature.

We redefined and re-calculated the work recovery efficiency (η) according to the literature in the revised manuscript. The changes mainly involve two aspects. First, the stress-strain recovery curve of the material is used to calculate the ideal consumed energy. Second, the repeated calculation of the energy loss of the motor in the real consumed energy of single-unit devices is deducted.

Additionally, we added more discussions on the correlations between the hysteresis of material and input work required in the system. To the best of our knowledge, the work recovery performance for the polymer system is firstly reported in this work.

The corresponding changes are as follows:

(1) Text

The work recovery efficiency (η) of the double-unit system compared to the single-unit cooling device, with the same mass of refrigerant, can be estimated by the ratio of the real recovery energy (W_r) to the ideal maximum recoverable work (W_i).³⁴ Considering a complete stretch-recovery cycle of a refrigerant sample, we obtain the following expression for η :

$$\begin{aligned}\eta &= \frac{W_r}{W_i} \times 100\% = \frac{(2W_{r,s} - 2W_{r,m}) - (W_{r,d} - W_{r,m})}{W_i} \times 100\% \\ &= \frac{2W_{r,s} - W_{r,d} - W_{r,m}}{W_i} \times 100\% \quad \text{Supplementary Equation (18)}\end{aligned}$$

where, $W_{r,s}$ and $W_{r,d}$ are the real consumed energy for single-unit and double-unit devices during a t_{cycle} respectively. $W_{r,m}$ is the energy loss caused by the motor within a rotation process. W_i is the ideal maximum recoverable work for a complete stretch-recovery cycle of a refrigerant sample. $W_{r,s} = 4.02$ J and $W_{r,d} = 4.14$ J can be obtained by multiplying the average power (Supplementary Fig. 18a, 18b) of the single-unit and double-unit device by t_{cycle} , respectively. $W_{r,m} = 2.39$ J can be obtained by multiplying the average power of the no-load motor by t_{cycle} (Supplementary Fig. 18c). $W_{r,m}$ is considered constant for both the single-unit and double-unit devices. In the double-unit system, two refrigerant samples are engaged in the operational process during a rotation of the motor throughout the t_{cycle} . Throughout this period, the two refrigerant samples undergo elongation and contraction, respectively. Consequently, $W_{r,d}$ can be conceptualized as the sum of the work consumed by a refrigerant sample through a complete stretch-recovery cycle and the energy loss (equal to $W_{r,m}$) of the motor during a single rotation. In the single-unit system, a refrigerant sample is engaged in the operational process during twice rotations of the motor throughout the $2t_{\text{cycle}}$. Throughout this period, a refrigerant sample undergoes one complete stretch-recovery cycle. Consequently, $2W_{r,s}$ can be conceptualized as

the sum of the work consumed by a refrigerant sample through a complete stretch-recovery cycle and the energy loss (equal to $2W_{r,m}$) of the motor during twice rotations. Comparing these two mechanisms, $W_r = 1.51$ J (the numerator of Supplementary Equation (18)) is the energy consumption difference of the refrigerant samples under these two kinds of systems, where both mechanisms drive a refrigerant sample to undergo a complete stretch-recovery cycle. This consumption difference could be considered as the real recovered energy generated by the double-unit operating mechanism. On the other hand, W_i could be considered from the material level. As illustrated in Supplementary Fig. 20, when assuming that the unloading work of the refrigerant could be completely reused, the integrated area of the mechanical hysteresis (red-shaded area) is the energy consumed per unit volume of the refrigerant for a complete stretch-recovery cycle. This consumption represents the loss of work at the material level, which is caused by the material properties (internal friction). In this case, the integral area of the recovery curve (blue-shaded area) can be considered as the maximum energy that can be recovered by the system during a stretch-recovery cycle for a unit volume of refrigerant. At the strain range from 100% to 500%, the ideal maximum recoverable work (denominator in Supplementary Equation (18), W_i) can be obtained as $W_i = 2.68$ J by multiplying the integral area of the blue shaded in Supplementary Fig. 20. According to Supplementary Equation (18), $\eta = 56.3\%$ of the double-unit cooling system under stable cycling can be obtained.

(Page 17-19 in the Supplementary Information)

(2) Text

Moreover, a large-deformation cooling system with a work recovery efficiency of 56.3% has been demonstrated.

(Line 11 of Page 2 in the Main Manuscript)

(3) Text

According to Supplementary Equation (18) and Supplementary Fig. 20, the work recovery efficiency of the double-unit device is calculated to as high as 56.3%.

(Line 11 of Page 13 in the Main Manuscript)

(4) Text

Moreover, the designed double-unit cooling system which adapted to the large-deformation characteristic of polymers not only possesses a high COP_{sys} of 8.3 but also presents an impressive work recovery efficiency of 56.3%.

(Line 7 of Page 14 in the Main Manuscript)

(5) Text

On the other hand, solid-state cooling critically depends on functional development of the cooling devices that are tailored to solid refrigerant materials and their responses to the external fields.^{14-15, 32-37} However, the inherent large-deformation characteristics of polymers limit the development of their *e*-CE cooling devices due to the demand for large driving strokes. Only a few works have recently been reported on the large-deformation cooling polymer systems.^{21, 22, 29, 38} For example, by harnessing snap-through instability in the soft capacitor to drive the expansion and contraction of an NR membrane balloon, a high operating frequency function was achieved for the *e*-CE cooling device.²¹ Another type of large-deformation cooling device utilizes the geometric design of NR tubes to endow the device with a thermal conductivity compensation function.^{22, 38} Nevertheless, the energy recovery has not yet been realized in the large-deformation system, which is crucial for further improving the system efficiency.³⁹

(Line 4 of Page 4 in the Main Manuscript)

(6) Figure

(Page 40 in the Supplementary Information)

(7) Figure

(Page 42 in the Supplementary Information)

(8) Reference

32. Defay, E., et al. Enhanced electrocaloric efficiency via energy recovery. *Nat. Commun.* **9**, 1827 (2018).
33. Li, J., et al. High cooling performance in a double-loop electrocaloric heat pump. *Science* **382**, 801-805 (2023).
34. Cui, H., et al. Flexible microfluidic electrocaloric cooling capillary tube with giant specific device cooling power density. *Joule* **6**, 258-268 (2022).
35. Ahčin, Ž., et al. High-performance cooling and heat pumping based on fatigue-resistant elastocaloric effect in compression. *Joule* **6**, 2338-2357 (2022).
36. Bruederlin, F., Ossmer, H., Wendler, F., Miyazaki, S. & Kohl, M. SMA foil-based elastocaloric cooling: from material behavior to device engineering. *J. Phys. D Appl. Phys.* **50**, 424003 (2017).
37. Kirsch, S. M. et al. NiTi-based elastocaloric cooling on the macroscale: From basic concepts to realization. *Energy Technol.* **6**, 1567–1587 (2018).
38. Sebald, G., et al. High-performance polymer-based regenerative elastocaloric cooler. *Appl. Therm. Eng.* **223**, 120016 (2023).
39. Qian, S. Thermodynamics of elastocaloric cooling and heat pump cycles. *Appl. Therm. Eng.* **219**, 119540 (2023).

(Page 17-18 in the Main Manuscript)

(9) Reference

34. Kabirifar, P. et al. Elastocaloric cooling: State-of-the-art and future challenges in designing regenerative elastocaloric devices. *J. Mech. Eng.* **65**, 615–630 (2019).

(Page 53 in the Supplementary Information)

REVIEWERS' COMMENTS

Reviewer #1 (Remarks to the Author):

Dear authors,

thank you for taking into account my input and addressing every question and suggestion.

I believe, that the revised manuscript is of very high quality and ready for publication.

Congratulations on this interesting work.

Reviewer #2 (Remarks to the Author):

The authors have satisfactorily revised the manuscript. I recommend its publication in Nature Communications. I look forward to their future refrigerant heat exchange and cycling frequency studies.

Reviewer #3 (Remarks to the Author):

I would like to congratulate the authors of this article on all their hard work. Overall, the authors have responded satisfactorily to my comments. I recommend to accept this article.

Response to reviewers' comments

Shearo-caloric effect enhances elastocaloric responses in polymer composites for solid-state cooling (NCOMMS-24-01724A)

Dear Reviewers:

We appreciate you for providing positive feedback and a thorough evaluation of our work.
We appreciate you for accept this work for publication.

Best regards

Reviewer #1 (Remarks to the Author)

Dear authors,

thank you for taking into account my input and adressing every question and suggestion.

I believe, that the revised manuscript is of very high quality and ready for publication.

Congratulations on this interesting work.

Response: We thank you so much for the thorough review and comments on our study!

Reviewer #2 (Remarks to the Author)

The authors have satisfactorily revised the manuscript. I recommend its publication in Nature Communications. I look forward to their future refrigerant heat exchange and cycling frequency studies.

Response: We appreciate your thorough review and precise comments on our study!

Reviewer #3 (Remarks to the Author)

I would like to congratulate the authors of this article on all their hard work. Overall, the authors have responded satisfactorily to my comments. I recommend to accept this article.

Response: We appreciate your affirmation and valuable feedback of this study!